# Development and Application of an Intelligent Assessment System for Mathematics Learning Strategy among High School Students—Take Jianzha County as an Example

**Guangming Wang [1], Xia Chen [1,\*], Dongli Zhang [1,\*], Yueyuan Kang [1], Fang Wang [1] and Mingyu Su [2]**

[1]  Faculty of Education, Tianjin Normal University, Tianjin 300387, China
[2]  School of Mathematical Sciences, Beijing Normal University, Beijing 100875, China
\*  Correspondence: 1801030028@stu.tjnu.edu.cn (X.C.); 20201089@qhnu.edu.cn (D.Z.)

**Abstract:** To improve the quality of mathematics learning for high school students in economically disadvantaged areas, and promote education equity and sustainable development, this study developed the Mathematics Learning Strategies Intelligent Assessment and Strategy Implementation System by using artificial intelligence technology. The system fuses assessment scales, a set of norms, improvement strategies, and the intelligent assessment and strategy implementation program into an organic whole. The system can intelligently output all participants' diagnosis results of the mathematics learning strategy in batches and automatically propose targeted improvement strategies for every participant. By applying the intelligent system to Jianzha County, Huangnan Tibetan Autonomous Prefecture, Qinghai Province, China, the results show that the mathematical learning strategies of high school students in Jianzha County were at a middle level; mathematical cognitive strategies and mathematical resource-management strategies need to be improved. The system's effectiveness in practical applications was later tested via both quantitative methods, such as questionnaire surveys and testing, and qualitative methods, such as interviews, as well as evaluation by self and others. By intervening with participants according to the strategy implementation program provided by the system, it was found that their mathematics learning strategy level improved. The results of the study show that the system can accurately diagnose the level of mathematics learning strategies of high school students and that interventions based on the improvement measures can improve students' mathematics learning strategy and mathematics achievements, indicating that the system is effective.

**Keywords:** mathematics learning strategies; intelligent assessment and strategy implementation system; high school students; intelligent diagnosis; improvement strategy; mathematics achievements



## 1. Introduction

Learning strategy is the sum of a series of scientific plans, rules, and sequences of actions developed by learners to improve their learning efficiency [1,2]. It not only contains intrinsic, implicit psychological control mechanisms, but also can be presented through external, explicit learning methods, learning behaviors, and operational processes, so it is often used to measure and evaluate whether students "know how to learn". Studies have shown that there is a significant positive correlation between learning strategies and academic performance, and learning strategies have a significant impact on students' learning efficiency [3–5]. Mathematical learning strategies are strategies that contribute to mathematical learning, including understanding, memorizing, applying, and solving mathematical problems of concepts, formulas, and so on [6]. Many factors influence children's mathematics learning during their growth [7–9], and mathematics learning strategies are one of the important influencing factors [4,10,11]. Effective mathematics learning strategies can significantly improve students' performance in mathematics and enhance their attitudes and emotions in mathematics learning [12,13]. Research on the assessment and intervention of mathematics learning strategy can help improve mathematics

learning efficiency, enable students to know how to learn, and promote sustainable learning and development [14,15].

According to the requirements outlined in the Fourteenth Five-Year Plan and the Outline of Vision 2035 of China, the education quality and education level in ethnic areas and economically less-developed areas will be improved. The "three districts and three prefectures" are deeply impoverished areas at the national level, among which Jianzha County, Huangnan Tibetan Autonomous Prefecture, Qinghai Province, China, is a key education-poverty-alleviation county and is also the current counterpart of Tianjin Normal University. Poverty alleviation through education is the fundamental strategy to block the intergenerational transmission of poverty [16]. Therefore, to help the poverty-stricken areas, efforts must be made to help the children living there receive a good education.

As research on mathematics learning strategy continues to evolve, many influential assessment tools have emerged [15,17]. However, most of these assessment tools can only be used in the classroom through paper-and-pencil tests or online questionnaires. Researchers often spend long hours collecting data, analyzing them, and preparing diagnostic reports. This task is not only labor intensive but also requires researchers to have the knowledge of statistical analysis and the ability to analyze data. In addition, most of the applied studies on mathematics learning strategy only suggest general recommendations on how to improve mathematics learning strategy based on the research results, but there are fewer individualized improvement strategies for students with different mathematics learning strategy levels, and there are even fewer projects that use artificial intelligence to develop a student learning monitoring system. Integrating artificial intelligence technology and developing an intelligent assessment system for mathematics learning strategies will help to provide new ideas for the "professional construction" of mathematics learning assessment in the new era.

With the development of artificial intelligence technology, intelligent diagnostic tools are more precise and effective than traditional assessment tools in measuring and evaluating mathematical learning strategies. Therefore, it is urgent to develop an intelligent assessment tool for mathematical learning strategies to apply it in high schools and then test the effectiveness of its application. This study aims to develop an intelligent assessment and strategy implementation system for mathematics learning strategies by integrating AI technology and intelligently evaluating and diagnosing mathematics learning strategies of students from different schools and different regions, at any time anywhere. This system can also intelligently prescribe "prescriptions" based on the "cause of the disease" so that researchers can conduct targeted intervention studies. We will apply this system to measure the level of mathematics learning strategies of high school students in Jianzha County, Huangnan Tibetan Autonomous Prefecture, Qinghai Province, China. The system can smart output diagnosis results reports for all participants in batches within a very short period, making the assessment process more efficient and more timely. At the same time, this system can intelligently prescribe "prescriptions" based on the "cause of the disease"; in other words, an intelligent diagnostic service is automatically provided for each participant student, i.e., "one person, one strategy", so that researchers can conduct targeted intervention studies. The system is of great significance to improving students' mathematics achievements and promoting the sustainability of their mathematics learning.

## 2. Literature Review

### 2.1. The Relationship between Learning Strategies and Academic Performance

Learning strategies are behaviors and thoughts affecting the learners' motivation or affective state, or how the learner selects acquires, organizes, and integrates new knowledge (Weinstein and Mayer, 1986). Learning strategies may be critical to academic performance [6], and are considered an essential variable in students' learning process [18,19]. Mastery learning strategies (MLS) are more effective than many established instructional approaches in promoting students' academic achievement (Bloom, 1976; Block and Burns, 1976). Learning strategies are often used to measure and assess "whether students can

learn". Research shows that there is a significant positive correlation between learning strategies and academic performance, and learning strategies can promote students' learning and improve students' academic achievement [18,20]. When learning mathematics, students typically acquire learning strategies through the guidance of teachers or peers to improve their learning efficiency. Students with outstanding mathematics performance typically adopt learning strategies appropriately to manage their learning [18,21]. By contrast, students with relatively poor mathematics performance typically cannot apply effective learning strategies to solve problems or monitor their learning [22–24]. Therefore, to realize the learning objectives, in the implementation of teaching and learning activities, teachers should pay attention to student learning styles and learning strategies so that learning achievement becomes optimal. Although prior research has investigated the relationship between individual learning strategies and mathematics achievements [15,25,26], there is less research investigating how integrations of learning strategies interact to foster students' sustainable learning.

*2.2. Assessment Tool for Mathematics Learning Strategies*

To improve students' learning strategies, it is first necessary to conduct a comprehensive assessment and systematic diagnosis of students' learning strategies. Therefore, a set of assessment tools for learning strategy with good measurement properties can provide a scientific basis for evaluation and ensure the accuracy of diagnosis.

There has been a long history of academic research on the development of assessment tools for middle school students learning strategies [27]. Weinstein and Mayer (1986) developed the Learning and Study Strategies Inventory (LASSI) for college students [28]. In 1990, Weinstein and colleagues adapted LASSI based on the learning characteristics of high school students and formed what is now called the Learning and Study Strategies Inventory for High School Version (LASSI-HS) [29]. LASSI-HS can comprehensively measure high school students' learning strategies and has become one of the few influential learning strategies assessment tools for high school students, and its effectiveness has been confirmed by many researchers. Stevens used a group of British and American subjects and another group of Mexican and American subjects to study and compare the three-factor model of LASSI-HS, and obtained evidence of factor invariance, proving that LASSI-HS is a multidimensional and Diagnostic measuring tool. Olivarez used confirmatory factor analysis to verify the construct validity of LASSI-HS. The reliability test showed that the reliability coefficient of internal consistency was 0.94. Murphy et al. also demonstrated the good measurement properties of the cross-cultural reliability and validity of LASSI-HS using Singaporean subjects; similarly, Samuelstuen also proved the reliability and validity of LASSI-HS in Norway.

However, it has been more than 20 years since LASSI-HS was published, and many topics are no longer applicable to today's students. Zhou Y. (2003) compiled the "Questionnaire on Learning Strategies for Junior High School Students", with a total of 56 questions including 4 aspects: cognitive strategy, meta-cognitive strategy, goal management, and planning strategy. These questionnaires are relatively complete, but the subjects are college students or junior high school students, so they cannot be directly applied to high school students [30]. Researchers used the LASSI-HS as a framework to revise a Chinese version of a learning strategy scale suitable for Chinese high school students, which contains 28 items [31]. Qimeng Liu et al. examined over 48,000 eighth-grade Chinese students' mathematics performance and use of learning strategies, exploring how the use of learning strategies and their combinations are related to students' mathematics performance in the Chinese context [15]. Based on the above research, we compiled a series of questionnaires on mathematics learning strategies, involving three stages primary school, middle school, and high school. Based on the existing mathematics learning strategy scale, empirical research is carried out in many places by establishing regional norms [32]. Among them, the "Questionnaire on Mathematical Learning Strategies of Senior High School Students" includes

three main dimensions Mathematical Cognitive Strategy, Mathematical Metacognitive Strategy, Mathematical Resource-management Strategy, and eleven sub-dimensions [33].

In the context of the information society, using modern information technologies such as artificial intelligence to explore how to evaluate students' learning has become a new requirement. Wang and colleagues independently developed the "Mathematics Learning Quality Intelligent Evaluation System", which can intelligently diagnose individual students on a large scale and automatically and efficiently output diagnosis reports; based on the diagnosed problems, the system automatically outputs targeted improvement measures.

*2.3. Intervention Studies to Improve Mathematics Learning Strategies*

It is generally believed that mathematical learning strategies can be trained. Similar to knowledge learning, the training of mathematics learning strategies requires long-term and step-by-step guidance to achieve the desired results. Oxford and Nyikos (1989) believe that the teaching of learning strategies needs to go through three processes: awareness training, metacognitive training, and specific strategy training, to effectively promote the internalization and transfer of students' strategies [34]. Studies on learning strategies in the last decade focused on interventions in learning strategies. Cho and Heron (2015) investigated students' self-regulated learning concerning affective aspects by using different learning strategies [3]. Among the recent studies on learning strategies, Gasco and colleagues (2014) compared the learning strategies of eighth- and ninth-grade students in mathematics in Basque Country (Spain) [35]. Cao Yiming and Chen Pengju (2018) used a mathematics learning strategy questionnaire and a mathematics ability assessment tool to obtain data from 1782 secondary school students in a region. The students were categorized into four ability levels, A, B, C, and D, based on their mathematical ability values. The study investigated the use of mathematical learning strategies and their effects on secondary school students at different ability levels and found that: the use of mathematical learning strategies by secondary school students was related to their ability levels, i.e., the higher-level students used strategies better than the lower-level students; the learning strategies of A and D levels were significantly different from other levels. Levels B and C differed only in terms of cognitive strategies [36]. Douglas J. Hacker conducted a metacognitive instructional intervention that was designed using the Self-Regulated Strategy Development model (SRSD) for teaching foundational concepts of fractions. The intervention was administered to upper elementary students who exhibit mathematics difficulties in computational accuracy, quality of mathematical reasoning, and number of rhetorical elements, and students' performance in these areas improved after the intervention [37].

Although research on the intervention of mathematics learning strategies varies among scholars, it all generally follows the following guiding principles: the principle of necessity, i.e., raising students' awareness of the application of learning strategies and making them realize the close connection between strategies and learning activities; the principle of subjectivity, i.e., giving full play to the teacher's guiding and modeling role to make students clear about the nature and use of strategies; and the principle of transferability, i.e., training students to choose appropriate learning strategies according to different learning contexts and different learning tasks.

Given the background of the information society, it has become a new era requirement to use modern information technology such as artificial intelligence to evaluate students' learning situations. Although researchers have developed a series of measurement and evaluation tools for mathematics learning strategies, few people use information technology and artificial intelligence technology to develop learning strategy assessment tools. In the process of assessment, diagnosis, and intervention, if the assessment and improvement report are manually generated for each student one by one, the workload is huge, the efficiency is low, and the coverage is small. In addition, the current research on teaching and guidance of learning strategies is almost for all students, lacking targeted and personalized guidance. Moreover, there is no unified system and method for the guidance of mathematics learning strategies, which needs further research by researchers.

In summary, although there has been much research on mathematics learning strategies, the research on the measurement and evaluation of mathematics learning strategies for high school students is not systematic. There is still a lack of representative regional norms, a lack of intervention research on improving high school students' mathematics learning strategies, and a lack of an intelligent evaluation application system.

*2.4. Research Questions*

The purpose of this study is to develop an intelligence assessment system of mathematics learning strategy for high school students to conduct an empirical study by implementing and applying it in Qinghai Province, China, in order to obtain the current status of mathematics learning strategy levels in high school students and summarize the characteristics of the mathematics learning strategy levels of high school students in this region compared with those in educationally developed regions of China (e.g., Tianjin City). We then test the effectiveness of its application with a case intervention study for students with poor assessment results.

The research questions of this study include three areas:

1. How to develop an intelligent assessment system of mathematics learning strategy for high school students?
2. How to apply the intelligent assessment system in Jianza County, Huangnan Tibetan Autonomous Prefecture, Qinghai Province, China?
3. How to test the effectiveness of its application?

## 3. Methodology

*3.1. Research Design*

Firstly, based on the theoretical orientation of mathematics learning strategies, we revised and established the "Mathematics Learning Strategies Questionnaire for Senior High School Students". Given that there is no norm of mathematics learning strategies for high school students in Jianzha County, we chose the norm of mathematics learning strategies for high school students in Tianjin as a reference frame. Tianjin, as one of the municipalities directly under the central government of China, is one of the most developed regions in China in terms of basic education. By comparing with Tianjin, we can find the gap in the education level between Jianzha County and the educationally developed areas. Based on the questionnaires and norms, this paper proposes a set of improvement strategies aiming at promoting high school students' mathematics learning strategies. Additionally, then, by using artificial intelligence technology to integrate assessment scales, norms, and strategies, we developed the scientific and logically sound Mathematics Learning Strategies Intelligent Assessment and Strategy Implementation System.

Secondly, the study applied the Mathematics Learning Strategies Intelligent Assessment and Strategy Implementation System in Jianzha county. On one hand, the system smart outputs the mathematics learning strategy diagnosis results of all high school students participating in the survey in batches; on the other hand, the system smart outputs the mathematics learning strategy improvement strategies for every participant. Through norm-referenced analysis, the study obtained the current status of mathematics learning strategy levels of high school students and summarized the characteristics of the mathematics learning strategy levels of high school students in this region compared with the educationally developed city Tianjin.

Lastly, case students with relatively low scores in mathematics learning strategies were selected for the intervention study. Additionally, then, the study conducted focused instruction and intervention for the case students for 3 months to test the applicative efficacy of the intelligent assessment system in both quantitative and qualitative terms. The roadmap of the research design is shown in Figure 1.

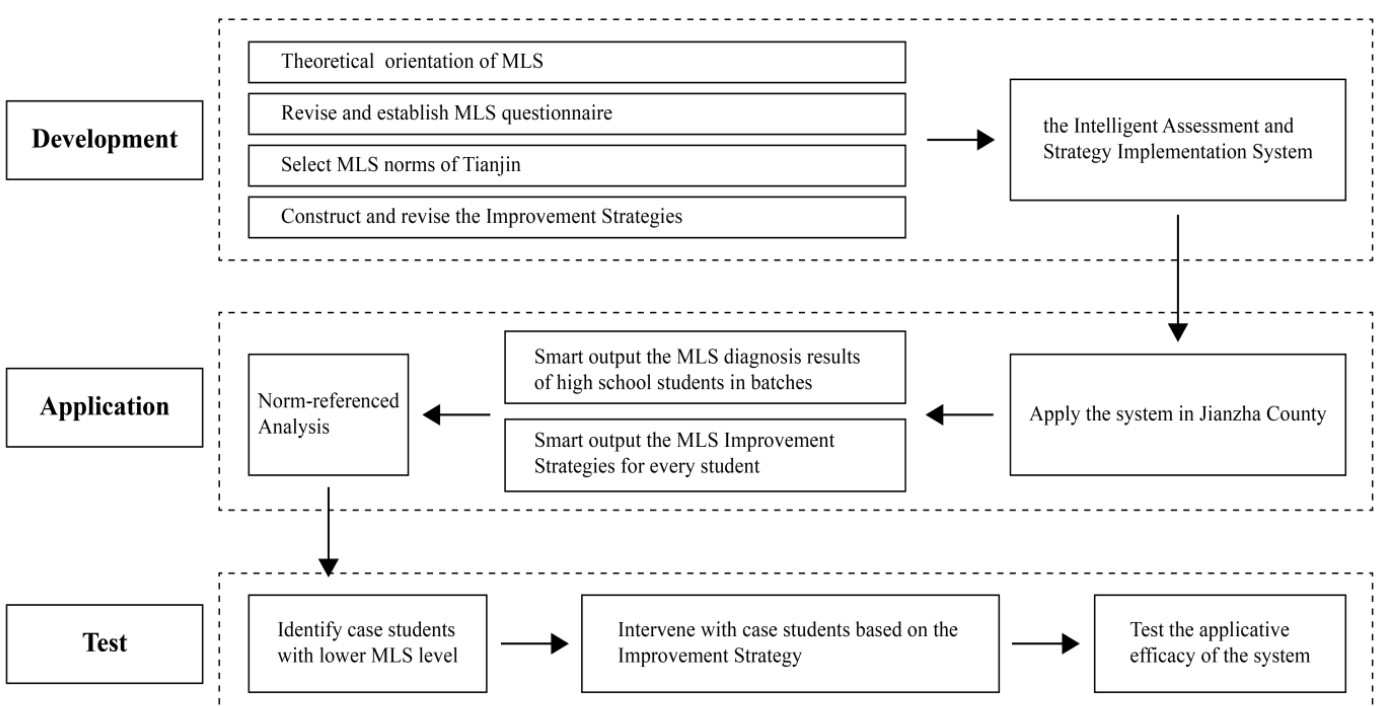

**Figure 1.** Research design on the development and application of the Mathematics Learning Strategies Intelligent Assessment and Strategy Implementation System.

### 3.2. Instruments

### 3.2.1. Mathematics Learning Strategies Questionnaire

According to the content of the study and the actual situation of the research subjects, we adjusted the "Mathematics Learning Strategies Questionnaire for Senior High School Students" [33] that we compiled independently. Several experts were consulted and the questionnaire was finalized after several tests and corrections. The adjusted scale (see Appendix A) contains a total of 54 questions, including a total of 20 questions on the mathematical cognitive strategy, 15 questions on the mathematical metacognitive strategy, 14 questions on the mathematical resource-management strategy, and polygraph and background information questions. The questions with * are reverse-scoring questions, as shown in Table 1.

By analyzing the valid data, it can be seen from Table 2 that the Cronbach's alpha coefficient of the total questionnaire is 0.946, and the coefficients of each dimension are around 0.8 and above; the Brown Spearman split-half reliability of the total questionnaire is 0.930, and the split-half reliability of each dimension is around 0.8 and above, which indicates that the reliability of the questionnaire is good.

**Table 1.** Distribution of Questionnaire Questions.

| Main Dimension | Sub-Dimension | Question Number | Amount |
|---|---|---|---|
| Mathematical cognitive strategy (MLS) | Retelling strategy (RS) | 8 *, 28, 40, 46, 47 | 5 |
| | Finishing strategy (FS) | 1, 3, 7, 11, 13 | 5 |
| | Organizing strategy (OS) | 4, 41, 45, 49, 51 | 5 |
| | Feedback strategy (FS) | 17, 23 *, 26, 39, 54 | 5 |
| Mathematical metacognitive strategy (MMS) | Planning strategy (PS) | 5, 14, 19, 33, 53 * | 5 |
| | Monitoring strategy (MS) | 2, 9, 10, 16, 24 | 5 |
| | Reflection and adjustment strategy (RAAS) | 20, 21 *, 42, 44, 52 | 5 |
| Mathematical resource-management strategy (MRMS) | Time-management strategy (TMS) | 22, 32, 48 | 3 |
| | Environmental-management strategy (EMS) | 15, 25 *, 30, 35 | 4 |
| | Mood-management strategy (MMS) | 27, 34, 37, 50 | 4 |
| | External help strategy (EHS) | 36, 38, 43 | 3 |
| Polygraph Questions | | 31–53, 9–29 | 4 |
| Background questions | | 6, 12, 18 | 3 |

Note: Questions with * are reverse-scoring questions.

**Table 2.** Reliability indicators of the questionnaire data.

| Dimension | Cronbach's a Coefficient | Split-Half Reliability |
|---|---|---|
| Mathematics Learning Strategies | 0.946 | 0.930 |
| Mathematical cognitive strategy | 0.894 | 0.883 |
| Mathematical metacognitive strategy | 0.881 | 0.871 |
| Mathematical resource-management strategy | 0.768 | 0.809 |

It can be seen from Table 3 that there is a significant correlation between the dimensions of the questionnaire, and the correlation coefficients are between 0.7 and 0.9, indicating that there is an independent relationship between the dimensions of the questionnaire. The correlation coefficient between each dimension and the overall questionnaire is above 0.9, which is greater than the correlation coefficient between each dimension, indicating that each dimension is not only relatively independent, but also has a certain contribution to the whole, so the overall structural validity of the questionnaire is good.

**Table 3.** Validity indicators of the questionnaire data.

| Dimension | MLS | MCS | MMS | MRMS |
|---|---|---|---|---|
| MLS | 1 | | | |
| MCS | 0.964 ** | 1 | | |
| MMS | 0.946 ** | 0.882 ** | 1 | |
| MRMS | 0.916 ** | 0.821 ** | 0.793 ** | 1 |

Note: ** indicates a significant correlation at the 0.01 level.

The questionnaire has a solid theoretical foundation, a reasonable structural framework, and good reliability and validity, so it can be used as a valid instrument to measure the level of mathematical learning strategies of high school students in Jianzha County, Qinghai Province; for the pre-test of mathematical learning strategies of intervention cases, it can also be used as a valid scale for the post-test of intervention.

### 3.2.2. Test of Mathematics Academic Achievement

Before the intervention, the study collected the subjects' monthly exam results, midterm exam results, final exam results, and class ranking of the three results (40 students in total) in the second semester of the 2020–2021 school year, and all three exam papers were assigned by members of the mathematics teaching and research team at the participants' school. After the intervention, the mathematics learning results were selected from the monthly exams, midterm exams, final exams in the first semester of the 2021–2022 school year, and the class ranking of the three results.

### 3.2.3. Intelligent Assessment System

This study used "The Intelligent Batch Assessment of Mathematics Learning Quality for Primary and Secondary School Students—Learning Strategies" Intelligent Assessment and Strategy Implementation System, which was independently developed by our team and obtained a national patent in China [38].

The intelligent system automatically calculates the total score of the main dimension and the scores of the sub-dimensions according to the results of the students' questionnaires, which can be viewed on the corresponding pages. The system can batch calculate the scores and grades of each tested student in the overall mathematics learning strategy and the three main dimensions, and import the diagnosis report into the specified folder. Based on this, researchers can identify typical cases with poor scores in each dimension of mathematics learning strategy, and then intervene in cases based on the improvement plan automatically derived by the system. After the intervention, the intelligent system is used to compare and analyze the diagnostic reports before and to test the effectiveness of the intelligent intervention.

### 3.3. Sample and Data Collection

We selected Jianzha County, a city in northwestern China, Qinghai Province, to conduct a study on the application and effectiveness testing of the intelligent assessment system. Since the students in the first year of high school have mastered some mathematics knowledge and are not under the pressure of the college entrance examination, we surveyed 223 first-year students of two schools in Jianzha County through a questionnaire survey. We obtained 195 valid questionnaires (84 boys, 98 girls, and 13 missing data), and the recovery rate was 87.44%.

With the assistance of mathematics teachers, we selected three students who were relatively weak in cognitive strategies, metacognitive strategies, and resource-management strategies, and applied the improvement strategies provided by the intelligent assessment and strategy implementation system to conduct individual key interventions on them. Their math scores were in the bottom 15% of recent exams, with long hours of study and high math stress. The students' mathematics scores are provided by the schools tested, and their study time and mathematics learning stress assessment are based on the level of all students in the school.

The research adopts a combination of qualitative and quantitative methods to examine the effectiveness of the application of the intelligent assessment system. Additionally, after the intervention, using the interview method, the participants and their mathematics teachers were invited to conduct interviews, so that the effect of the intervention was observed from an external evaluation.

### 3.4. Intelligent Intervention Process

Step 1: Intelligent assessment to understand the diagnosis results. Through the diagnostic results of the intelligent system of mathematics learning strategies, the individual can understand their problems and obtain the suggestions provided by the intelligent system. These suggestions can make students realize the practical significance of mathematics learning strategies in the process of mathematics learning practice.

Step 2: Explain the instructions and detail the learning strategies. Explain clearly to students the content of each dimension of the mathematics learning strategy during the intervention, and combine the operational definitions of each dimension with examples of how the strategy can be used in the learning of mathematics subjects, so that students can appreciate the close connection between the strategy and their mathematics learning lives and improve their understanding of mathematics learning strategies.

Step 3: Demonstrate the application of strategies. According to the plan proposed by the intelligent assessment system, the strategy content is divided into several dimensions, and the learning of strategies is divided into several small steps. Additionally, then, the teacher will demonstrate and explain how to apply the strategy in the mathematics learning tasks.

Step 4: Students self-explain the strategy content. Students will try to combine the strategy content with the learning points and express them in words through the teacher's demonstration and explanation. In the presentation process, attention should be paid to retelling key content. Teachers should prompt and correct them promptly. Moreover, they must give affirmation to students after they have completed their retelling.

Step 5: Apply learning strategies. Teachers guide students to try to apply strategies in learning activities. When students apply strategies, teachers should provide supervision or hints and help students who encounter problems. Teachers should give feedback on positive mathematical learning behaviors in individual cases and remind them of their mistakes in terms of poor learning behaviors.

Step 6: Transfer learning strategies. Teachers should communicate with students promptly during the intervention period to understand the use of strategies, students' awareness and level of applying strategies, and guide students to transfer learning strategies in their daily learning activities.

Step 7: Intelligent testing of strategy levels. After three months of supervision and instruction, students' effectiveness in using the strategies should be diagnosed again using an assessment tool (questionnaire).

Step 8: Generalize and improve learning strategies. Students are informed of the test results to improve their confidence in applying the strategies and promote their application and generalization in their daily learning activities.

## 4. Results

### 4.1. Development of Intelligent Assessment System

The intelligent assessment system for mathematics learning strategy of high school students mainly included assessment models (structural model, assessment index system), assessment scale, regional norms, targeted improvement strategies for students at different levels, and intelligent assessment software that intelligently presented the above module contents (see Figure 2).

Since 2012, the research team has begun to develop a series of mathematics learning quality measurement tools for elementary, middle, and high school students on metacognition, non-intellectual factors, and learning strategies. Additionally, by applying measurement tools in practice, the team has developed regional norms of mathematics learning quality, such as mathematics learning strategy for students of different grades in multiple districts in Tianjin, China. Thirdly, based on the diagnostic results of the assessment, individualized improvement strategies have been developed and provided for students at different levels of mathematics learning strategy. Finally, an intelligent assessment software was developed, through which participants can quickly obtain assessment and diagnosis results and suggestions of how to improve their learning strategies. The whole process is realized intelligently and efficiently by using computer software. The above four contents together constitute the Mathematics Learning Strategies Intelligent Assessment and Strategy Implementation System. The specific research and development process and main content of each module in the system are as follows.

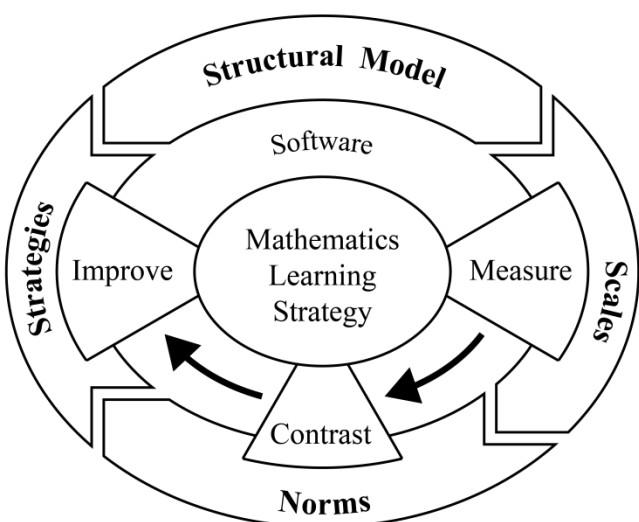

**Figure 2.** Intelligent assessment system and strategy implementation for mathematics learning strategy of high school students.

### 4.1.1. Theoretical Framework and Assessment Scale

By referring to a large number of studies, we developed the "Questionnaire on Mathematics Learning Strategies for Senior High School Students". The scale has good reliability and validity. Using the CNKI database as the data source, as of 24 March 2022, the questionnaire has been downloaded 2402 times, cited 93 times, and has been applied in more than 140 schools in Beijing, Tianjin, Chongqing, Shandong Province, etc. The research background of this questionnaire is Chinese high school students, which is more consistent with the research background of this study. According to the characteristics of high school students' mathematics learning in Jianzha County, we tested and revised the scale many times, and finally determined the questions of the questionnaire. The questionnaire includes 3 main dimensions (mathematical cognitive strategy, mathematical metacognitive strategy, mathematical resource-management strategy) and 11 sub-dimensions (retelling strategy, finishing strategy, organizing strategy, feedback strategy, planning strategy, monitoring strategy, reflection and adjustment strategy, time-management strategy, environmental-management strategy, mood-management strategy, external help strategy) (see Figure 3).

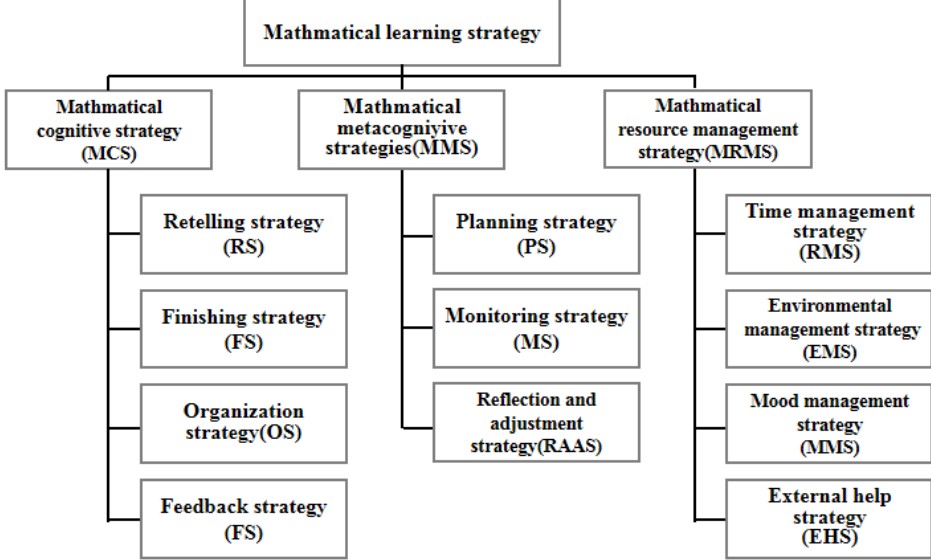

**Figure 3.** Structural model of mathematics learning strategy questionnaire.

4.1.2. Regional Norms

To better understand the current situation of high school students' mathematics learning strategies in Jianzha County, not only authoritative measurement tools are needed, but also reliable evaluation standards should be selected to make the results of the current situation analysis more convincing. To understand the current situation of math learning strategy levels of high school students in Jianzha County, we need not only authoritative measurement tools, but also reliable evaluation criteria to make the results of the current situation analysis more convincing.

Our research team sampled a large number of high school students in Tianjin and established "The Norm of High School Students' Mathematics Learning Strategies" and their level standards (their level rating criteria). The regional norms in Tianjin classified the overall level of mathematics learning strategies of high school students into five levels: "excellent, upper–middle, middle, lower–middle, and poor".

The development level of basic education in Tianjin is currently better than that of Qinghai Province, and the norm of mathematics learning strategies for high school students in Qinghai Province has not been established. Given the background of the Chinese government's steady promotion of high-quality and balanced development of basic education, taking the norm of mathematics learning strategies for high school students in Tianjin as a yardstick to compare the current situation of high school students' mathematics learning strategies in Jianzha County will help to find the gap between Jianzha County and the educationally developed areas, and discover the weakness of Jianzha County's high school students in mathematics learning to provide them with more accurate assistance for basic education.

Two types of norms were selected: percentile rank norms and standard score norms, so that the original scores could be fully explained. The general norms of mathematics learning strategy, the norms of the three main dimensions of mathematical cognitive strategy, mathematical metacognitive strategy, mathematical resource-management strategy, and 10 sub-dimensions were established, and the grade evaluation criteria were also established (see Table 4).

**Table 4.** Grade of mathematics learning strategy and its sub-dimensions of high school students in Tianjin.

|  | Level | T-Score | Raw Score (X) | Percentage Rating (PR) |
|---|---|---|---|---|
| MLS | excellent | T ≥ 68 | X ≥ 213 | PR ≥ 96.49 |
|  | upper–middle | 68 > T ≥ 56 | 213 > X ≥ 176 | 96.49 > PR ≥ 73.78 |
|  | middle | 56 > T ≥ 44 | 176 > X ≥ 148 | 73.78 > PR ≥ 27.39 |
|  | lower–middle | 44 > T ≥ 32 | 148 > X ≥ 125 | 27.39 > PR ≥ 3.61 |
|  | poor | 32 > T | 125 > X | 3.61 > PR |
| MCS | excellent | T ≥ 68 | X ≥ 88 | PR ≥ 96.49 |
|  | upper–middle | 68 > T ≥ 56 | 88 > X ≥ 73 | 96.49 > PR ≥ 74.46 |
|  | middle | 56 > T ≥ 44 | 73 > X ≥ 61 | 74.46 > PR ≥ 29.43 |
|  | lower–middle | 44 > T ≥ 32 | 61 > X ≥ 50 | 29.43 > PR ≥ 3.61 |
|  | poor | 32 > T | 50 > X | 3.61 > PR |
| MMS | excellent | T ≥ 68 | X ≥ 67 | PR ≥ 96.49 |
|  | upper–middle | 68 > T ≥ 56 | 67 > X ≥ 61 | 96.49 > PR ≥ 75.05 |
|  | middle | 56 > T ≥ 44 | 61 > X ≥ 56 | 75.05 > PR ≥ 33.53 |
|  | lower–middle | 44 > T ≥ 32 | 56 > X ≥ 39 | 33.53 > PR ≥ 4.68 |
|  | poor | 32 > T | 39 > X | 4.68 > PR |
| MRMS | excellent | T ≥ 68 | X ≥ 59 | PR ≥ 96.59 |
|  | upper–middle | 68 > T ≥ 56 | 59 > X ≥ 50 | 96.59 > PR ≥ 76.51 |
|  | middle | 56 > T ≥ 44 | 50 > X ≥ 43 | 76.51 > PR ≥ 37.33 |
|  | lower–middle | 44 > T ≥ 32 | 43 > X ≥ 35 | 37.33 > PR ≥ 4.00 |
|  | poor | 32 > T | 35 > X | 4.00 > PR |

### 4.1.3. Development of Improvement Strategy

Based on the normative criteria and the operational definition of mathematics learning strategies, the "Countermeasures for improving the mathematics learning strategy of high school students" was proposed after several rounds of revisions. (See Table 5)

**Table 5.** Countermeasures for improving the mathematics learning strategy of high school students.

| Performance | Improvement Strategy |
| --- | --- |
| **Dimension: Mathematical Cognitive Strategy (MCS)** | |
| Sub-dimension: Retelling strategy (RS) | |
| Middle-level student ($10 \leq X < 20$): There is still room for improvement in learning through review, practice, etc. Low-level student ($X < 10$): There are deficiencies in learning mathematics through review and practice. | **1. Review at the right time, review in time, and review often** Ebbinghaus' law of forgetting points outlines that forgetting begins immediately after learning, and the forgetting speed is fast at first, and then gradually decreases over time. To achieve the best memory effect, the review should be carried out as soon as possible after the study, and the review content and time should be reasonably arranged to improve review efficiency. The content of daily review should be appropriate, so as not to be too tense or too fatiguing and so as not to affect the review effect. **2. Participate in memory with multiple senses** The simultaneous participation of eyes, ears, hands, and mouth in the process of memorization can effectively improve the efficiency of memory. For example, listening while watching, talking while writing, listening while doing, and thinking while doing after-school examples and exercises. **3. Diversify the forms of practice to prevent negative emotions such as boredom due to monotonous practice forms.** Diversified forms of practice are conducive to the better consolidation of knowledge, enhancing learning effects, and do not easily lead to boredom. For example, when reviewing mathematical concepts, formulas, theorems and algorithms, you can read them aloud, copy them, write them down, use them to solve examples in books and exercises after class, and ask and answer each other among classmates in various ways. |
| Sub-dimension: Finishing strategy (FS) | |
| Middle-level students ($10 \leq X < 20$): 1. There is room for improvement in analyzing and processing learning materials in depth and detail and in understanding the deeper meaning inherent in them. 2. There is still room for improvement in deepening mathematical understanding by establishing connections between old and new knowledge. Low-level students ($X < 10$): 1. There are deficiencies in analyzing and processing learning materials in depth and detail and in understanding the deeper meanings inherent in them. 2. There are deficiencies in deepening mathematical understanding by making connections between old and new knowledge. | **1. Use mnemonics to improve the effectiveness of memorization** In the process of learning mathematics, use the mnemonics taught by the teacher to assist in the memorization of knowledge. For example, in trigonometric functions, you can use the phrase "odd change and even change unchanged, symbols look at the quadrant" to aid memory. **2. Develop the habit of taking mathematical notes to promote understanding of knowledge** By taking notes, you can establish connections between old and new knowledge and promote understanding of knowledge. When taking notes, keep the following in mind: (1) Grasp the key points and make comments during the lecture; (2) Use mind maps, circle annotations, etc., to enrich the content of notes; (3) Add examples and classic question types under the properties of the corresponding theorems; (4) Read the notes at any time, and constantly revise and improve the notes. **3. Try to explain the mathematical concepts and other knowledge in your language to deepen the understanding and memory of knowledge.** After a series of studies such as previews and lectures, you will have a certain understanding of the mathematical knowledge you have learned. You can further try to express the results of your understanding of knowledge and the process of thinking in your language. This is better than mere memorization. For example, after learning about "sets", use your language to describe the definition and related properties of sets. |

**Table 5.** *Cont.*

| Performance | Improvement Strategy |
|---|---|
| | Sub-dimension: Organizing strategy (OS) |
| Middle-level students ($10 \leq X < 20$):<br>1. There is still room for improvement in integrating the internal connection between old and new knowledge;<br>2. There is still room for improvement in refining the knowledge learned and constructing it to form a new knowledge structure.<br>Low-level students ($X < 10$):<br>1. There are deficiencies in integrating the internal connection between old and new knowledge;<br>2. There are deficienciesin refining the knowledge learned and constructing it to form a new knowledge structure. | **1. Regularly summarize the knowledge points and framework of a module or unit by using outlines, tables, graphics, etc.**<br>Present the learned mathematical concepts, principles, formulas and other knowledge in the form of diagrams or outlines, which not only helps to understand the connection between various knowledge points, but also facilitates memory and application. It can be presented as follows:<br>(1) Outlines<br>Write out the main points of knowledge. For example, since the vector is something that students have not been exposed to, and is highly abstract, the knowledge points are relatively fragmented. Therefore, you need to list the key knowledge in the textbook in the form of an outline to promote understanding and memory.<br>(2) Tables<br>In the general review, the form of tables can be used more often to compare the knowledge. For example, when learning the equation of a line, because their equation forms are easily confused, you can organize them with the help of tables.<br>(3) Graphs<br>Explain the connection between knowledge with the help of graphs. For example, a flow chart can be used to summarize the steps of solving a problem, which helps to improve the overall grasp of knowledge and the ability to solve problems. For example, when performing three-dimensional geometry, first build the system, then write the coordinates and vectors, and finally bring the formula.<br>**2. Categorize and organize the questions you have formulated, integrate the solutions and refine them**<br>You can prepare a notebook to organize the questions you have asked, and annotate the knowledge points involved with eye-catching colors or marks next to the questions. Sort out and integrate the examination points and solutions of the same type of questions, and refine the methods of solving the questions so that you can apply them flexibly and learn from them.<br>**3. Regularly classify and organize what you have learned**<br>Regularly classify the knowledge you have learned according to the logical relationship, and organize it according to your thinking habits, so that you can comprehensively understand all the contents of the knowledge you have learned, and also facilitate the extraction of knowledge. For example, after learning "point, line and surface", you can organize the three parts by combining them in pairs. |
| | Sub-dimension: Feedbacking strategy (FS) |
| Middle-level students<br>($10 \leq X < 20$):<br>It is easier for them to adjust their attitude and methods of mathematics learning through the external assessment and diagnosis of their mathematics learning situation;<br>Low-level students ($X < 10$):<br>It is difficult for them to adjust their mathematics learning attitude and methods through the external assessment and diagnosis of their mathematics learning situation. | **1. Make timely use of the teacher's feedback in the classroom**<br>Pay attention to the teacher's feedback, such as the teacher's comments and follow-up questions given to your own or other students' answers; an affirmative smile and gesture from the teacher. In this way, you can obtain encouragement or find out the shortcomings of your own or your classmates' thinking, and then gain experience in mathematical learning activities, so that your mathematical ability can be further developed at the original level and enhance your self-confidence in mathematics learning.<br>**2. Use timely feedback from homework and test papers to analyze and adjust study attitudes and methods independently**<br>By analyzing the feedback of the homework and test papers, you can effectively understand your learning situation, find deficiencies and adjust your learning attitude and methods in time, to improve the current situation of learning and improve learning efficiency. For example, the teacher's judgment on the correctness of your homework, the marking at a certain place, or the comments.<br>Think about and correct your mistakes promptly, and then reflect on how you master the mathematical knowledge content covered by the homework, and whether your attitude towards mathematics learning in the recent period is correct and whether the method of mathematics learning is appropriate. |

| Performance | Improvement Strategy |
|---|---|
| | **3. Actively test your mastery of knowledge through exercises** <br> By reflecting on the exercises, you can effectively understand your mastery of knowledge. For example, if you make a mistake in an exercise, you should revise it in time and think about what type of question is being examined. Which aspect of knowledge is being examined? Why did I make a mistake? At the same time, reflect on the process of solving the problem and the method of solving it, and summarize your experience to improve your knowledge and learning ability. |
| **Dimension: Mathematical metacognitive strategy (MMS)** | |
| **Sub-dimension: Planning strategy (PS)** | |
| Middle-level students ($10 \leq X < 20$): <br> 1. There is room for improvement in planning the various activities involved in completing the task, predicting the results, selecting strategies, envisioning ways to solve problems, and estimating their effectiveness before the learning activity begins; <br> 2. There is room for improvement in clarifying activity goals before learning activities begin. <br> Low-level students ($X < 10$): <br> 1. There are deficiencies in planning the various activities involved in completing the task, predicting the results, selecting strategies, envisioning ways to solve the problem, and predicting its effectiveness before the learning activity begins; <br> 2. There are deficiencies in clarifying the objectives of the learning activities before they begin. | **1. Set reasonable and specific short-term and long-term goals for mathematics learning based on your current mathematics learning status (situations)** <br> Mathematics learning goals play a role in guiding, motivating and regulating the mathematics learning process. By breaking down long-term goals into short-term goals, you can overcome the difficulties and challenges you encounter in mathematics learning by accomplishing one short-term goal after another, and eventually reaching your long-term goals. For example, if you set a goal for yourself at the beginning of the semester "to achieve a good grade in mathematics at the end of the semester", you can break this long-term goal down into several short-term goals, such as paying attention and listening carefully in math class every day, completing homework assignments in a timely and quality manner, and reading the textbook or looking up your notes in time when it comes to mathematical knowledge you have forgotten or not mastered. <br> **2. Formulate a clear mathematics learning plan, according to the mathematics learning objectives** <br> A mathematics learning plan can promote and regulate mathematics learning, make mathematics learning more organized, and thus improve the efficiency of mathematics learning. Make a clear plan for each stage of mathematics learning according to your specific situation. For example, make a semester plan for mathematics learning at the beginning of the semester; make a daily plan for mathematics learning after school. In general, you can review first and then answer the questions to consolidate the mathematical knowledge learned that day, master the basic questions covered by the content, and then preview the mathematical knowledge to be learned in the next class, and clarify the key points, difficulties, doubts, and confusion points. Finally, summarize and evaluate the completion of the plan promptly is completed well, you can give yourself a small reward to satisfy the sense of achievement; if the plan is poorly completed, you can give yourself a certain small punishment. <br> **3. Arrange the test time reasonably during the math test** <br> When you receive the papers for the exam, you should have an overview of the papers, including the type of questions and the number of questions, and arrange the order and time of the questions reasonably. When encountering a problem that cannot be solved, learn to temporarily give up or use other methods to help solve it. After answering all the questions that can be answered in the test paper, think about the questions that cannot be answered during the inspection process that are guaranteed to obtain all the points. Make sure that you receive all the marks you can. |
| **Sub-dimension: Monitoring strategy (MS)** | |
| Middle-level students ($10 \leq X < 20$): <br> 1. There is room for improvement in testing the learning process promptly and the learning objectives. <br> 2. There is room for improvement in finding the gaps between learning objectives or plans and current learning activities to successfully achieve effective mathematical learning. | **1. When solving mathematical problems, compare the results obtained with life experience and learning experience** <br> Abstracting mathematical knowledge in real life can arouse students' enthusiasm for learning and deepen students' understanding of knowledge. For example, when learning plane geometry, you can think about the positional relationship between two straight lines, the positional relationship between lines and surfaces, and the positional relationship between surfaces and surfaces by looking for examples in life, such as corners, buildings, etc. |

**Table 5.** *Cont.*

| Performance | Improvement Strategy |
| --- | --- |
| Low-level students (X < 10):<br>1. There are deficiencies in testing the learning process according to the learning objectives;<br>2. There are deficiencies in finding the gaps between learning objectives or plans and current learning activities to successfully achieve effective mathematical learning. | **2. Develop the habit of checking results**<br>After completing the homework or test paper, you need to check whether the solution result is correct. You can solve the problem with different ideas to see if you can obtain the same answer, bring the answer into the question for verification, or recalculate the question. When recalculating, pay attention to overcoming the influence of inertial thinking, which means rereading the question, trying not to be disturbed by the thinking of the first time you did the problem.<br>**3. Narrate problem-solving ideas by "thinking aloud"**<br>"Thinking aloud" is to express the whole process of thinking in your language and make it "externalized". For example, in the process of solving the problem, identify the theorems and conditions on which your problem-solving is based; state the goal of the problem and the plan to achieve the goal; evaluate the advantages and disadvantages of your problem-solving thinking process; discover and describe the different ideas that may appear in the problem-solving process. These are all conducive to actively mobilizing the role of self-monitoring in problem-solving by improving the accuracy of problem-solving and improving the level of mathematical thinking. |

| Sub-dimension: Reflection and adjustment strategy (RAAS) | |
| --- | --- |
| Middle-level students (10 ≤ X < 20):<br>1. There is still room for improvement in identifying deviations in the mathematics learning process based on the results of monitoring learning activities, and adjust learning strategies or revise learning goals in a timely manner.<br>2. There is still room for improvement in evaluating the learning results at the end of the mathematics learning activities, correct your mathematics learning behaviors, and remedy deficiencies.<br>Low-level students (X < 10):<br>1. There are deficiencies in identifying deviations in the mathematics learning process based on the results of monitoring learning activities, and adjust learning strategies or revise learning goals.<br>2. There are deficiencies in evaluating the learning results at the end of the mathematics learning activities, correct your mathematics learning behaviors, and remedy deficiencies. | **1. Always reflect on your learning attitude and state**<br>You should always reflect on whether you can pay attention in class, whether you can control your behavior effectively when studying, whether you have the confidence to overcome learning difficulties, and what methods should be used to solve them in the future if there are deficiencies. Try to propose corresponding goals, summarize promptly and reflect on your progress and deficiencies.<br>**2. Correct attribution of learning results**<br>Mathematics performance is affected by many factors, so you need to understand it correctly and rationally. From a subjective point of view, your efforts are strongly associated with learning results. In general, attributing learning results to subjective factors will increase your motivation to learn and, when learning results are poor, you will not be less inclined to study harder because of it. From an objective point of view, academic performance is also associated with the difficulty of test questions, luck, physical and mental state, and external environment, so attributing learning results to objective factors can help you to relieve anxiety to some extent.<br>**3. Deepen thinking and understanding of learning through self-questioning**<br>You will have deep thinking and understanding of mathematical problems by constantly asking yourself questions, thus improving your mathematical learning ability and learning effectiveness. You should often ask yourself "did I solve this question this way?" "Could it be easier to do it differently?" "Why did I do it wrong?" |

| **Dimension: Mathematical resource-management strategy (MRMS)** | |
| --- | --- |
| Sub-dimension: Time-management strategy (RMS) | |
| Middle-level students (6 ≤ X < 12):<br>1. There is still room for improvement in the overall arrangement of time;<br>2. There is still room for improvement in the efficient use of optimal mathematics learning time;<br>3. There is still room for improvement in the flexible use of fragmented time to learn mathematics. | **1. List the things that need to be performed and classify them**<br>List the things that need to be performed on a list of activities, sort and classify them according to priority: the important and urgent things, the important and not urgent things, and the unimportant and not urgent things. Prioritize the important and urgent tasks, and then perform the important and non-urgent tasks, and ask your family and classmates to help you with the unimportant and urgent tasks. The unimportant and non-urgent tasks will not take much time. |

**Table 5.** *Cont.*

| Performance | Improvement Strategy |
|---|---|
| Low-level students (X < 6):<br>1. There are deficiencies in the overall arrangement of time;<br>2. There are deficiencies in the efficient use of optimal mathematics learning time;<br>3. There are deficiencies in the flexible use of fragmented time to learn mathematics. | **2. Make the most of your prime study time and study the most important content when you are at your best**<br>The same hour of work can make a significant difference in learning efficiency depending on the mental state of the individual. The key to effective learning is to use the most efficient study time wisely and to make good use of the golden hours of the day, such as in the morning and before going to bed. In addition to receiving enough rest, it is also important to enjoy a good amount of physical exercise to maintain the best study condition. In addition, whenever you feel tired, you should stop for a short break, drink a glass of water or take a walk, and then concentrate on your studies.<br>**3. Strictly set the time to finish homework to avoid procrastination**<br>Before doing homework each time, set the time required to complete the homework according to the amount of homework and your actual situation, and then complete the homework on time within the specified time, which will help to cultivate your time concept. For example, you can make a detailed homework schedule for yourself. The content of the schedule includes what homework to write in a certain period and how much to write, and then completing the homework on time within the time period specified in the homework plan. |
| Sub-dimension: Environmental-management strategy (EMS) | |
| Middle-level students (8 ≤ X < 16):<br>1. There is room for improvement in attributing mathematics learning outcomes correctly.<br>2. There is still room for improvement in adjusting the bad mood of mathematics learning.<br>3. There is still room for improvement in promoting mathematics learning through perseverance and self-reinforcement.<br>Low-level students (X < 8):<br>1. There are deficiencies in attributing mathematics learning outcomes correctly.<br>2. There are deficiencies in adjusting the bad mood of mathematics learning;<br>3. There are deficiencies in promoting mathematics learning by perseverance and self-reinforcing. | **1. Cultivate a positive mindset in learning mathematics through self-encouragement**<br>Encouraging yourself frequently can help to improve self-confidence, reverse the bad mindset of mathematics learning, give full play to your inner potential, and thus improve the self-awareness of learning. In the process of learning mathematics, you should always encourage yourself with affirmative and motivational words and phrases, including positive psychological suggestions such as telling yourself "I am capable of learning mathematics well".<br>**2. Correctly face the bad emotions in mathematics learning**<br>When bad emotions arise in the process of mathematics learning, you can divert your attention by taking a short break, taking a deep breath, or doing some other healthy activities that you are interested in. In addition, you should also appreciate the mathematical principles in your daily life by heart, appreciate the beauty and value of mathematics, cultivate and stimulate good motivation to learn mathematics, and actively regulate and rid yourself of bad emotions.<br>**3. Develop good willpower such as self-control and persistence**<br>Learning mathematics is a long-term, complex mental task. Perseverance and independence are essential qualities for successful mathematics learning activities. We can sharpen our will in the process of solving difficulties, for example, by doing some difficult problems to cultivate our perseverance and not giving up easily in the face of difficulties; we can also use our spare time to read mathematics extracurricular books and learn the rigorous scientific attitude and hard work of mathematical scientists. |
| Sub-dimension: Mood-management strategy (MMS) | |
| Middle-level students (8 ≤ X < 16):<br>1. There is still room for improvement in finding favorable conditions to assist mathematics learning;<br>2. There is still room for improvement in storing orderly mathematical materials, planning the learning space, and creating a mathematical learning atmosphere. | **1. Create an ideal learning environment**<br>An ideal learning environment can facilitate your efficient learning, on the one hand, and cultivate your good study habits on the other hand. For example, pay attention to the orderly arrangement of desks, chairs and books; regularly organize mathematics study materials such as mathematics papers, mathematics study cases, and mathematics homework books after finishing the mathematics study of a certain module or a certain unit.<br>**2. Establish a good teacher–student relationship and create a harmonious and pleasant atmosphere in the mathematics classroom**<br>A good teacher–student relationship helps to create a harmonious and pleasant atmosphere in mathematics classrooms, and can encourage you to learn mathematics happily in mathematics classrooms. Therefore, you must learn to respect and appreciate teachers, form a harmonious teacher–student relationship with teachers, and actively interact with them. |

**Table 5.** *Cont.*

| Performance | Improvement Strategy |
|---|---|
| Low-level students (X < 8):<br>1. There are deficiencies in finding favorable conditions to assist mathematics learning;<br>2. There are deficiencies in the orderly storage of mathematical materials, the planning of learning space, and the creation of a mathematical learning atmosphere. | **3. Choose an appropriate math learning environment**<br>A quiet and comfortable learning environment is conducive to your mathematics learning and thinking. On the contrary, a noisy environment will make you restless and inattentive, which is not conducive to normal study and life. Therefore, you should choose to study mathematics in a quiet environment with suitable light and temperature. |
| Sub-dimension: External help strategy (EHS) | |
| Middle-level students (6 ≤ X < 12):<br>1. There is still room for improvement in seeking help from others to overcome difficulties or solve problems when encountering difficulties in mathematics learning, to improve learning efficiency and achieve expected learning goals.<br>2. There is still room for improvement in choosing an appropriate time for help and seeking targeted help, when encountering difficulties or problems in mathematics learning.<br>Low-level students (X < 6):<br>1. There are deficiencies in seeking help from others to overcome difficulties or solve problems when encountering difficulties in mathematics learning.<br>2. There are deficiencies in choosing an appropriate time for help and seeking targeted help. | **1. Correct your attitude towards helping**<br>A good attitude will make your inquiry the best. Before asking for help, be clear about the questions you want to ask and find out what you do not understand. For example, read the topic first, understand the conditions in the topic and the questions asked, try to find ideas, and then seek help from others after no results.<br>**2. Choose the right person for help according to different problems**<br>Anyone who helps to solve the problem can become the object of academic help. Therefore, you should choose different help objects according to your own needs, such as classmates who you get along with well, teachers who know how to solve problems and can give you advice and guidance promptly, parents, books, network resources, etc.<br>**3. Identify the problem and choose the right time to ask for help**<br>Before seeking help from others, it is important to think about the problem, rather than asking for help immediately without thinking. Additionally, consider the time and emotional state of others, and do not ask for help when they are not available.<br>**4. Choose the appropriate way to communicate effectively with the object of assistance, so as to seek targeted help**<br>Effective communication with the person you are seeking help from is the key to attaining help effectively in the process of seeking help to solve the problem. You can choose the appropriate communication channel according to your actual situation, such as face-to-face, by telephone, online and so on. In the process of asking for help, you should express your problems in an orderly manner, so that the person you are asking for help can understand your problems and provide accurate help. |

### 4.1.4. Intelligent Assessment Software

The system is developed with the Microsoft Visual Studio Community 2019 tool and NPOI plug-in, combined with Sunny UI for interface beautification. (See Figure 4) Two intelligent pieces of assessment software were designed and developed for different scenarios, namely the Individual Student Edition and the Integrated School Edition. Both software packages include a complete high school math strategy assessment scale and a math strategy improvement response form, the latter of which provides professional assessments and individualized recommendations for students based on the assessment results. The difference is that the Individual Student Edition allows individual students to make their diagnoses and provides immediate diagnostic results and suggestions for improvement; the Integrated School Edition uses the school and district as a whole and outputs visual diagnostic results and suggestions for improvement for the district, school, class as a whole and all of its students within a short period.

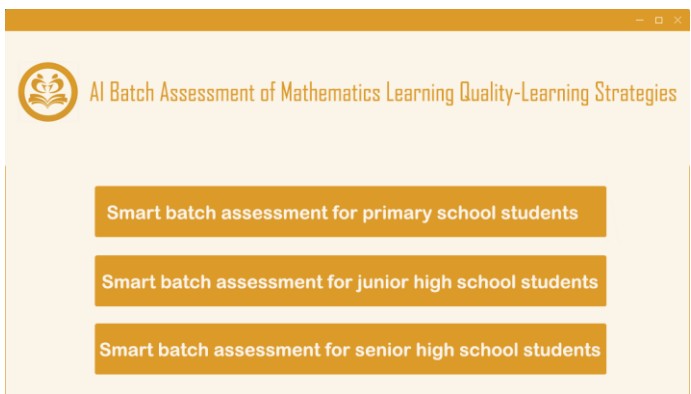

**Figure 4.** The main interface of mathematics learning strategy intelligent assessment and strategy implementation system.

On the one hand, students can independently and instantly view the diagnostic results after completing the test using the individual assessment and strategy software; at the same time, the system can automatically and instantly provide feedback on improvement measures according to the level of diagnostic results, making it convenient for students to independently view the results anytime and anywhere. The picture below shows the case of a student who used the system to independently assess the level of mathematics learning strategies, and received immediate feedback and suggestions for improvement. See Figure 5.

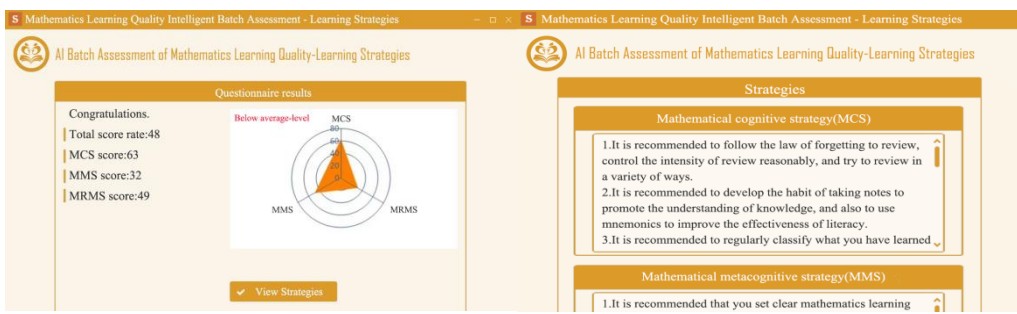

**Figure 5.** Visualized diagnosis results of mathematics learning strategy of individual self-assessment and strategy implementation software.

On the other hand, the intelligent batch assessment and Strategy Implementation software can automatically calculate the total score of the main dimension and the score of the sub-dimension according to the results filled in by the students. (See Figure 6) The software is divided into three modules: the first is the basic information collection module, the second is the scale data collection module and the third is the results and recommendations output module. Assessors can view their scores and grade levels on different screens for the main dimension, sub-dimensions, total score level, sub-dimension score rate, and current performance, and can view targeted improvement measures based on their current level. The diagnostic results and improvement measures of the mathematics learning strategy can also be automatically exported in batches.

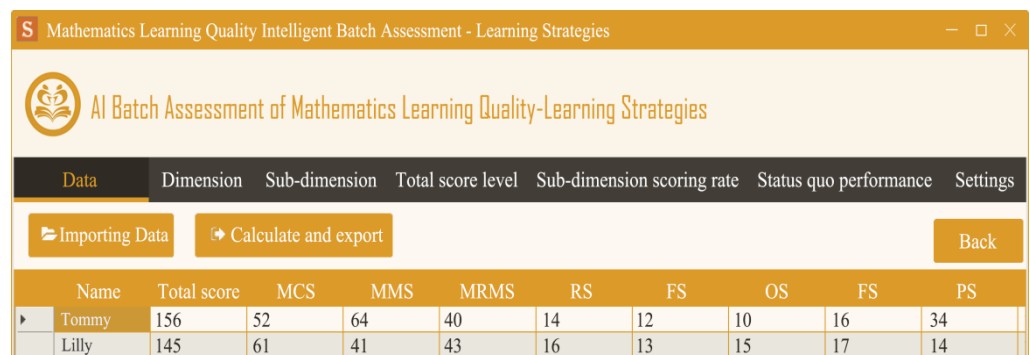

**Figure 6.** The functional interface of the software for mathematics learning strategy intelligence batch assessment and strategy implementation.

The Intelligent Assessment and Implementation system for mathematics learning strategies use information technology to accurately diagnose the level of students' mathematics learning strategies, enhance students' understanding of their mathematics learning strategies and improve their behaviors, thereby helping to improve students' mathematics learning efficiency.

*4.2. Intelligence Assessment Diagnosis and Norm-Referenced Analysis*

4.2.1. Diagnostic Results of Mathematics Learning Strategies and the Norm Reference Analysis of High School Students in Jianzha County

**(1) Diagnostic results of mathematics learning strategies of high school students in Jianzha County**

Descriptive statistics are carried out on students' mathematics learning strategies and primary dimensions (see Table 6), and the mean value of the questionnaire is 166.78 (out of 245 points). The scoring rate (scoring rate = mean/dimension full score, the higher the score rate, the more concentrated in the high segment, indicating that the students' mathematical metacognition level is higher) of mathematical learning strategies is 68.07%, indicating the overall situation of students' mathematics learning strategies is normal. Among them, the highest score is 241 points and the lowest score is 113 points. A pairwise *t*-test was performed on the three first-order dimensions, and the results showed that there was a significant difference between the three ($p = 0.000 < 0.01$), and the difference was large ($d > 0.8$). Specifically, the scoring rates of mathematics cognitive strategies and mathematics resource-management strategies were similar and lower than the overall scoring rate of the questionnaire, and only the mathematics metacognitive strategy (S = 69.20%) was higher than the overall scoring rate of the questionnaire.

**Table 6.** Descriptive statistics of mathematical learning strategies and the first-level dimensions.

|  | N | Min | Max | Mean (M) | Standard Deviation | Dimension Full Score | Score Rating (S) |
|---|---|---|---|---|---|---|---|
| Mathematics learning strategy | 195 | 113.0 | 241.00 | 166.78 | 25.38 | 245 | 68.07% |
| MCS | 195 | 42.00 | 97.00 | 67.66 | 10.61 | 100 | 67.66% |
| MMS | 195 | 32.00 | 74.00 | 51.90 | 8.41 | 75 | 69.20% |
| MRMS | 195 | 27.00 | 73.00 | 47.23 | 7.87 | 70 | 67.47% |

Among the mathematical cognitive strategies dimension, the scoring rates of the second-level dimension: Fine Processing Strategy, Retelling Strategy, Organization Strategy, and Feedback Strategy were 68.84%, 68.26%, 66.69%, and 66.85%, respectively. Among them, the scoring rates of Fine Processing Strategy and Retelling Strategy were higher than that of the first-level dimension mathematical cognitive strategies (S = 67.66%), while the

scoring rates of Organization Strategy and Feedback Strategy were lower than that of the first-level dimension mathematical cognitive strategies.

Among the mathematics metacognitive strategy dimension, the scoring rates of the second-level dimensions monitoring strategy, planning strategy, reflection strategy, and adjustment strategy were 71.14%, 67.57%, and 68.88%, respectively. Among them, only the scoring rate of the monitoring strategy was higher than that of the first-level dimension mathematics metacognitive strategy (S = 69.20%).

Among the mathematical resource-management strategies dimension, the scoring rates of the second-level dimensions time-management strategy, external help-seeking strategy, mindfulness strategy, and environmental-management strategy were 66.97%, 68.24%, 69.00%, and 65.72%, respectively. Among them, except for external help-seeking strategy and mindfulness strategy, the scoring rates of the other second-level dimensions were lower than that of the first-level dimension mathematical resource management strategy (S = 67.47%).

According to the survey on the level of mathematics learning strategies of high school students in Jianza County, Qinghai Province, their scores on the sub-dimensions were ranked as follows: Monitoring Strategies > mind management strategies > reflection and regulation strategies > refinement strategies > retelling strategies > external help strategies > planning strategies > time-management strategies > feedback strategies > organizational strategies > environmental-management strategies, indicating that the subjects' organizational and environmental-management strategies need to be focused on improving (See Table 7).

**Table 7.** Results of descriptive statistics of sub-dimensions of mathematics learning strategies.

| Dimensions | N | Min | Max | Mean (M) | Standard Deviation | Dimension Full Score | Score Rating (S) |
|---|---|---|---|---|---|---|---|
| Finishing Strategy | 195 | 8 | 25 | 17.21 | 3.23 | 25 | 68.84% |
| RS | 195 | 10 | 25 | 17.06 | 3.12 | 25 | 68.26% |
| OS | 195 | 9 | 25 | 16.67 | 3.29 | 25 | 66.69% |
| Feedback Strategy | 195 | 10 | 25 | 16.71 | 2.78 | 25 | 66.85% |
| MS | 195 | 9 | 25 | 17.78 | 3.17 | 25 | 71.14% |
| PS | 195 | 10 | 25 | 16.89 | 3.09 | 25 | 67.57% |
| RAAS | 195 | 9 | 25 | 17.22 | 3.07 | 25 | 68.88% |
| TMS | 195 | 4 | 15 | 10.05 | 3.01 | 15 | 66.97% |
| EHS | 195 | 5 | 15 | 10.24 | 2.12 | 15 | 68.24% |
| MMS | 195 | 7 | 20 | 13.80 | 2.62 | 20 | 69.00% |
| EMS | 195 | 7 | 20 | 13.14 | 2.51 | 20 | 65.72% |

**(2) Norm-referenced analysis to explore the level of students of Jianzha County in Tianjin norm**

The research team applied the existing "High School Students' Mathematical Learning Strategy Level Questionnaire" to conduct large-scale sampling among high school students in Tianjin to establish a regional norm in Tianjin. Two types of norms, the percentage-level norm and the standard-score norm, were selected so that the original scores could be fully explained. The overall norm of mathematics learning strategy, the norms of 3 main dimensions of mathematics cognitive strategy, mathematics metacognitive strategy, and mathematics resource-management strategy, and the norms of 11 sub-dimensions under the 3 main dimensions were established, respectively, and the grade evaluation standard was established, as shown in the Table 8.

**Table 8.** Grade of mathematics learning strategy and its sub-dimensions of high school students in Tianjin.

| | Level | T-score | Raw Score (X) | Percentage Rating (PR) |
|---|---|---|---|---|
| **MLS** | Top | T ≥ 68 | X ≥ 213 | PR ≥ 96.49% |
| | Above Average | 56 ≤ T < 68 | 176 ≤ X < 213 | 73.78% ≤ PR < 96.49% |
| | Average | 44 ≤ T < 56 | 148 ≤ X < 176 | 27.39% ≤ PR < 73.78% |
| | Below Average | 32 ≤ T < 44 | 125 ≤ X < 148 | 3.61% ≤ PR < 27.39% |
| | Low | T < 32 | X < 125 | PR < 3.61% |
| **MCS** | Top | T ≥ 68 | X ≥ 88 | PR ≥ 96.49% |
| | Above Average | 56 ≤ T < 68 | 73 ≤ X < 88 | 74.46% ≤ PR < 96.49% |
| | Average | 44 ≤ T < 56 | 61 ≤ X < 73 | 29.43% ≤ PR < 74.46% |
| | Below Average | 32 ≤ T < 44 | 50 ≤ X < 61 | 3.61% ≤ PR < 29.43% |
| | Low | T < 32 | X < 50 | PR < 3.61% |
| **MMS** | Top | T ≥ 68 | X ≥ 67 | PR ≥ 96.49% |
| | Above Average | 56 ≤ T < 68 | 61 ≤ X < 67 | 75.05% ≤ PR < 96.49% |
| | Average | 44 ≤ T < 56 | 56 ≤ X < 61 | 33.53% ≤ PR < 75.05% |
| | Below Average | 32 ≤ T < 44 | 39 ≤ X < 56 | 4.68% ≤ PR < 33.53% |
| | Low | T < 32 | X < 39 | PR < 4.68% |
| **MRMS** | Top | T ≥ 68 | X ≥ 59 | PR ≥ 96.59% |
| | Above Average | 56 ≤ T < 68 | 50 ≤ X < 59 | 76.51% ≤ PR < 96.59% |
| | Average | 44 ≤ T < 56 | 43 ≤ X < 50 | 37.33% ≤ PR < 76.51% |
| | Below Average | 32 ≤ T < 44 | 35 ≤ X < 43 | 4.00% ≤ PR < 37.33% |
| | Low | T < 32 | X < 35 | PR < 4.00% |

To further diagnose the level of mathematics learning strategies of high school students in Jianzha County, a norm of mathematics learning strategies for high school students in Tianjin (Wang, Li, and Sun, 2017) was applied to accurately diagnose the strengths and weaknesses of their mathematics learning strategies. After calculation, the average score of high school students' mathematics learning strategy is 166.78, and their percentage level is 40.35, indicating that their mathematics learning strategy level exceeds about 40.35% of high school students in Tianjin. It can be judged that the overall ability of their mathematics learning strategies is in the middle level of the Tianjin norm (see Table 8). The average score was evaluated by using the norms of three main dimensions in Tianjin. From the figure, it can be found that the students' mathematical cognitive strategies, mathematical metacognitive strategies, and mathematical resource-management strategies basically reached the middle level of the norm, and each dimension exceeded about 44.74%, 24.95%, and 39.57% of high school students in Tianjin, respectively. However, there is a certain gap with the excellent level of the norm, indicating that the students need to be improved in the three dimensions. It shows that students have deficiencies in many aspects, such as how to plan mathematics learning as a whole, how to reasonably choose specific learning content, how to allocate time, and how to scientifically arrange learning progress and other strategies. In addition, students failed to make timely changes in their behavior, psychology and learning styles, and methods when the state of mathematics learning and the mathematics learning environment changed during the mathematics learning process.

Based on the above analysis, high school students in Jianzha County have an average level of mathematics learning strategies in terms of the whole (See Figure 7), so in the next

intervention study, the focus should be on their resource-management strategies, with an emphasis on their time-management strategies and environmental-management strategies.

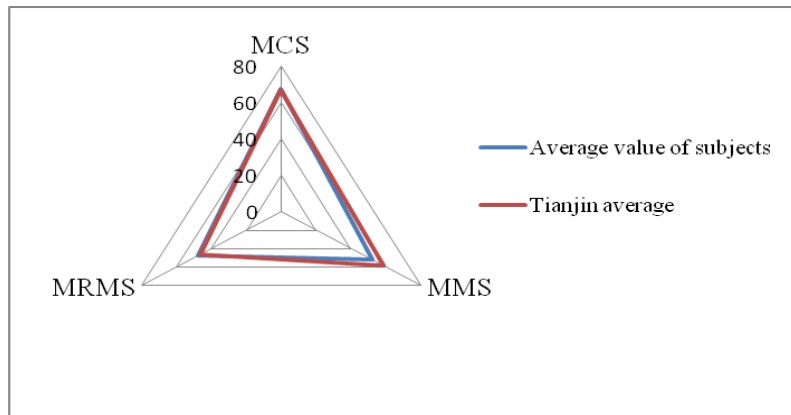

**Figure 7.** Comparison of the first-level dimension of mathematics learning strategies between the subjects and the Tianjin norm.

4.2.2. Diagnostic Results and Norm-Referenced Analysis of the Case Students' Mathematics Learning Strategies

According to the diagnosis results of the case class, after consulting teacher H, the head teacher and mathematics teacher of a class in the first grade of the X school, three students were selected for case study and diagnosis, using Student A, Student B, Student C as a proxy for their real information. The three students' mathematics learning strategies were at the lower–intermediate level or poor level. They spent a long time on mathematics learning, but their performance in mathematics was relatively poor, so they felt high stress in mathematics learning. It can be seen from Table 9 that the raw scores of mathematics metacognition of the three students are 116, 119, and 131, respectively, and the score rates are 47.35%, 48.57%, and 53.47%, respectively, which are in the lower–intermediate level or poor level of the norm.

**Table 9.** Results of norm-reference analysis of mathematics learning strategy in 3 students participating in the case study.

| Student | Raw Score | Norm Level | Scoring Rate (S) |
|---------|-----------|------------|------------------|
| A | 116 | Low | 47.35% |
| B | 119 | Low | 48.57% |
| C | 131 | below average | 53.47% |

In terms of cognitive strategies, both Student A and Student B were at a poor level, and Student C was at a lower–intermediate level. In terms of metacognitive strategies, both Student A and Student C were at the lower-intermediate level, and Student B was at the poor level. In terms of resource-management strategies, Student A and Student B were at the lower–intermediate level, and Student C was at the poor level. Therefore, Student A and Student B are weak in mathematical cognitive strategies, Student B is weak in mathematical metacognitive strategies, and Student C is weak in mathematical resource management. (See Table 10)

**Table 10.** Results of norm-reference analysis of sub-dimensions of mathematics learning strategy in 3 students participating in the case study.

| Sub-Dimensions | Student | Raw Score | Norm Level | Scoring Rate (S) |
|---|---|---|---|---|
| | A | 41 | Low ($X < 50$) | 41.00% |
| MCS | B | 49 | Low ($X < 50$) | 49.00% |
| | C | 57 | Below Average ($50 \leq X < 61$) | 57.00% |
| | A | 39 | Below Average ($39 \leq X < 56$) | 52.00% |
| MMS | B | 30 | Low ($X < 39$) | 40.00% |
| | C | 47 | Below Average ($39 \leq X < 56$) | 62.67% |
| | A | 36 | Below Average ($35 \leq X < 43$) | 51.43% |
| MRMS | B | 40 | Below Average ($35 \leq X < 43$) | 57.14% |
| | C | 27 | Low ($X < 35$) | 38.57% |

*4.3. Test of Applicative Efficacy of the Intelligence Assessment and Strategy Implementation System*

To further understand the details of case students in their mathematics learning, we conducted individualized in-depth interviews with the students A, B, and C, respectively. Please refer to Appendixes B–D for the specific outline of the interviews. After a 3-month focused instruction and intervention for individual students, post-tests were conducted on students' mathematics learning strategy levels and academic performance, combined with qualitative examinations of students' behavioral performance, students' self-assessment results, and teacher-assessment results, to test the applicative efficacy of the intelligent assessment system in both quantitative and qualitative terms.

4.3.1. Analysis of Applicative Efficacy on Student A with Weak Cognitive Strategies

**(1) Scale Analysis of the Improvement of Student A's Mathematical Learning Strategies**

According to the comparative study, following the mathematics learning strategies intervention, the post-test score of Student A was 156, which was 40 points higher than before the intervention, indicating the level of mathematics learning strategy improved significantly. In terms of cognitive strategy, Student A's score improved to 63, which was 54% higher than that before the intervention. In terms of individual changes in intervention, Student A's math learning strategies were at a poor level before the intervention, and Student A was at an intermediate level after the intervention.

In terms of cognitive strategies, Student A was at a poor level before the intervention, and Student A was at an intermediate level after the intervention. In conclusion, the above results indicate that the improvement strategy for the mathematics cognitive strategy dimension is effective, and the intervention is more effective when it is conducted based on the intelligent strategy-implementation plan. (See Table 11)

**Table 11.** Comparison of Student A's score before and after the intervention.

| | Student | Score Before Intervention | Score After Intervention |
|---|---|---|---|
| MLS | A | 116 | 156 |
| MCS | | 41 | 63 |

**(2) Analysis of effectiveness in terms of behavioral performance and academic achievement improvement of Student A**

After communicating with Student A's mathematics teacher about his behavioral performance, we learned that Student A's mathematics learning behavior had changed as follows: 1. He began to regularly organize the knowledge that he has learned and correct

the mistakes that he has made; 2. he was able to review what he has learned frequently, and use a variety of methods to memorize new formulas and theorems; 3. he began to think more actively about the connection between old and new knowledge; 4. when encountering unfamiliar math problems, he can try various methods to solve them and accumulate problem-solving experience; 5. he was able to accept the teacher's evaluation humbly, and adjust his learning style accordingly, to make continuous progress.

Referring to the chart, Student A's mathematics academic performance and class rank after the intervention were improved compared with those before the intervention. The *t*-test was carried out on the mathematics learning performance and class ranking of Student A before and after the intervention. The significance was less than 0.05, and there was a significant difference. Further observation of the effect size shows that the effect size of mathematics academic performance and class ranking before and after the intervention are greater than the large effect size ($d_1 = 0.850$, $d_2 = 0.985$) (The standard proposed by J. Cohen is adopted: 0.2 corresponds to a small effect size, 0.5 corresponds to a medium effect size, and 0.8 corresponds to a large effect size.), and the degree of variance was significant.

Specifically, after the intervention, Student A's mathematics scores and class ranking in the monthly, mid-term, and final exams improved compared to those before the intervention, and their ranking in the monthly and mid-term exams after the intervention gradually improved, except that their class ranking dropped two places at the final exam. (See Table 12)

**Table 12.** Student A's math scores and class rank before and after the intervention.

| | A's Math Score | | A's Class Rank | |
| --- | --- | --- | --- | --- |
| | Before Intervention | After Intervention | Before Intervention | After Intervention |
| monthly test scores | 80 | 92 | 40 | 32 |
| midterm exam results | 72 | 90 | 40 | 25 |
| final exam results | 69 | 91 | 40 | 27 |

**(3) Evaluation of the effectiveness of intelligent intervention for Student A**

Student A's math teacher mentioned, "Student A has made great changes in his math learning process and I noticed in the recent weekly test that he has done better in mastering the basic knowledge of mathematics than before. This intelligent system not only allows each student to recognize his or her problems but also allows the teacher to keep abreast of the level of mathematics learning strategies of each student in the class and also helps the teacher to intervene and guide the student in conjunction with the system's professional improvement measures". This demonstrates the applicative efficacy of the intelligence assessment and strategy implementation system in helping teachers assess and improve students' mathematics learning strategies.

Student A mentioned, "I found the recommendations in the report easy to understand and closely related to my mathematics learning life. According to the suggestions, I can consciously keep good habits, listen carefully in class, follow the teacher's rhythm and take notes, and also avoid walking away from my thoughts in class. I feel that the quality of my study has improved and I have made great progress in the midterm and final math exams". This demonstrates the applicative efficacy of the intelligence assessment and strategy implementation system in helping students self-assess and improve mathematics learning strategies.

4.3.2. Analysis of Applicative Efficacy on Student B with Weak Metacognitive Strategies

**(1) Scale Analysis of the Improvement of Student B's Mathematical Learning Strategies**

According to the comparative study, following the mathematics learning strategies intervention, the post-test score of Student B was 154, 35 points higher than before the intervention, indicating the level of mathematics learning strategies improved significantly. In terms of metacognitive strategies, Student B's score increased to 48, 60% higher than

that before the intervention. In terms of individual changes during the intervention, Student B's mathematics learning strategies were at a poor level before the intervention and at an intermediate level after the intervention. In terms of metacognitive strategies, Student B was at a poor level before the intervention and at a lower–intermediate level after the intervention. In conclusion, the above results about Student B indicate that the improvement strategy for the mathematics metacognitive strategy dimension is effective, and the intervention is more effective when it is conducted based on the intelligent strategy-implementation plan. (See Table 13)

**Table 13.** Comparison of Student A's score before and after the intervention.

|  | Student | Score Before Intervention | Score After Intervention |
|---|---|---|---|
| MLS | B | 119 | 154 |
| MMS |  | 30 | 48 |

**(2) Analysis of effectiveness in terms of behavioral performance and academic achievement improvement of Student B**

Through communication with B's mathematics teacher, we found that Student B's mathematics learning behavior had changed as follows: 1. He can make a plan before starting mathematics learning activities, and complete his learning tasks on time; 2. the phenomenon of inattentiveness in his mathematics class gradually decreases; 3. he gradually has the awareness of monitoring his speed while solving mathematics problems; 4. he can consciously analyze the reasons and can correct the deficiencies in time when there are ups and downs in mathematics performance; 5 he can learn from students with better mathematics performance, acknowledge their excellence in mathematics learning methods and mathematics learning habits, instead of simply thinking that they only have high IQ.

Referring to the chart, Student B's mathematics academic performance and class rank after the intervention were improved compared with those before the intervention. The *t*-test was carried out on the mathematics learning performance and class ranking of student B before and after the intervention, all showing significant differences, and the effect size was greater than the large effect ($d_1 = 0.923$, $d_2 = 0.943$), and the degree of variance was significant. Specifically, after the intervention, Student B's mathematics scores and class rank in the monthly exam, mid-term exam, and final exam were improved compared with those before the intervention. In the final exam, his class rank steadily improved. (See Table 14)

**Table 14.** Student B's math scores and class rank before and after the intervention.

|  | B's Math Score | | B's Class Rank | |
|---|---|---|---|---|
|  | Before Intervention | After Intervention | Before Intervention | After Intervention |
| monthly test scores | 85 | 95 | 35 | 29 |
| midterm exam results | 81 | 92 | 35 | 24 |
| final exam results | 79 | 93 | 36 | 24 |

**(3) Evaluation of the effectiveness of intelligent intervention for Student B**

Student B's math teacher mentioned: "Student B's math learning behavior has improved significantly, and he can consciously make demands on himself following the diagnostic recommendations, and his learning efficiency and learning quality have improved significantly compared to before. The intelligent system makes personalized suggestions for each student, which helps teachers to tailor their teaching to their students' needs". This demonstrates the applicative efficacy of the intelligence assessment and strategy implementation system in proposing time-efficient and personalized measures to address the weaknesses and promote the improvement of students' mathematics learning strategies.

Student B mentioned: "I have developed the habit of setting goals and plans for a mathematics study. Before, I knew that my learning efficiency was very poor, but I did not know how to improve my efficiency. Through this assessment, I have a clear understanding of my problems and I will follow the improvement measures in the report. Now I give myself a time limit to complete the homework before doing it and make a study plan for myself before I start each study task. After some time, I can feel that my mathematics learning efficiency has improved and my learning burden has been greatly reduced". We can see Student B can consciously apply mathematics learning strategies in daily mathematics learning; for example, he has developed the mathematics learning habit of setting goals and plans. He will follow the improvement measures on the diagnosis report to improve the mathematics learning strategy. This demonstrates the applicative efficacy of the intelligence assessment and strategy implementation system in improving students' efficiency towards mathematics learning and reducing their burden of mathematics learning.

4.3.3. Analysis of Applicative Efficacy on Student C with Weak Resource Management Strategies

**(1) Scale Analysis of the Improvement of Student C's Mathematical Learning Strategies**

According to the comparative study, following the mathematics learning strategies intervention, the post-test score of Student C was 152, 21 points higher than before the intervention, indicating the level of mathematics learning strategy improved significantly. In terms of resource-management strategy, Student C's score increased to 44, 63% higher than that before the intervention. In terms of individual changes during the intervention, Student C's mathematics learning strategies were at a lower–intermediate level before the intervention and at an intermediate level after the intervention. In terms of resource-management strategies, Student C was at a poor level before the intervention and at an intermediate level after the intervention. In conclusion, the above results about Student C indicate that the improvement strategy for the mathematics resource-management strategy dimension is effective, and the intervention is more effective when it is conducted based on the intelligent strategy-implementation plan. (See Table 15)

**Table 15.** Comparison of Student A's score before and after the intervention.

|  | **Student** | **Score Before Intervention** | **Score After Intervention** |
|---|---|---|---|
| MLS | | 131 | 152 |
| MRMS | C | 27 | 44 |

**(2) Analysis of effectiveness in terms of behavioral performance and academic achievement improvement of Student C**

Through communication with B's mathematics teacher, we found that Student B's mathematics learning behavior had changed as follows: He could start to use the fragmented time to correct mistakes and memorize formulas; he could consciously create a good mathematics learning environment and organize learning materials promptly to facilitate revision; he could encourage himself in the face of unsatisfactory mathematics results and maintain a positive learning attitude and self-confidence; he could think for himself and seek help from friends or teachers when encountering difficult problems.

Referring to the chart, Student B's mathematics academic performance and class rank after the intervention were improved compared with those before the intervention. The *t*-test was carried out on the mathematics learning performance and class ranking of Student C before and after the intervention. The *t*-test was carried out on the mathematics learning performance and class ranking of Student C before and after the intervention, both showing significant differences, and the effect size was greater than the large effect ($d_1 = 0.825$, $d_2 = 0.930$); the degree of variance was significant. Specifically, after the intervention, Student C's mathematics scores and class rank in the monthly exam, mid-term exam, and

final exam were improved compared with those before the intervention. The class ranking of Student C gradually improved in the three post-intervention exams. (See Table 16)

**Table 16.** Student B's math scores and class rank before and after the intervention.

| | C's Math Score | | C's Class Rank | |
|---|---|---|---|---|
| | **Before Intervention** | **After Intervention** | **Before Intervention** | **After Intervention** |
| monthly test scores | 91 | 98 | 33 | 26 |
| midterm exam results | 87 | 93 | 32 | 23 |
| final exam results | 85 | 97 | 34 | 21 |

**(3) Evaluation of the effectiveness of intelligent intervention for Student C**

The mathematics teacher of Student C mentioned: "Student C can consciously change himself following the recommendations on the diagnostic report in his daily mathematics learning. His learning behavior and learning enthusiasm have improved significantly compared with before, and his mathematics performance has also improved". This demonstrates the applicative efficacy of the intelligence assessment and strategy implementation system in helping students to find their weaknesses and effectively promote their mathematics learning strategies.

Student C mentioned: "Before it was always difficult for me to regulate the bad emotions in mathematics learning, so my enthusiasm for mathematics learning was relatively low. After three months of intervention guidance, I learned many methods to regulate bad emotions, Additionally, I can encourage myself from time to time. I was always embarrassed to ask my teacher or my classmates when I encountered difficulties in math learning. Now I can try to solve it with my classmates around me or ask my teacher for help. I feel better than before, my enthusiasm for mathematics learning has improved a lot, and I have also found a sense of achievement in mathematics learning". Student C has mastered a wealth of methods to regulate negative emotions after the intervention, which indicates the applicative efficacy of the intelligence assessment and strategy implementation system in regulating students' learning emotions and improving their enthusiasm for mathematics learning.

To sum up, a 3-month intervention was carried out for the case students using the intelligence assessment and strategy implementation system. Using a variety of methods such as tests, questionnaires, interviews, text analysis, etc., the study found that the case students improved in mathematics academic performance, mathematics learning strategy level, mathematics learning habits, mathematics learning behavior, etc. The three students and their teachers all reflect that the quality of mathematics learning has been improved and the learning burden has been reduced. This proves that the intelligence assessment and strategy implementation system is efficient, convenient, and scientific.

## 5. Discussion and Conclusions

The level of mathematical learning strategies of high school students in Jianzha County was obtained by an intelligent assessment and strategy implementation system, and the study showed that there are differences when students are using mathematical cognitive strategies, metacognitive strategies, and resource-management strategies. The results based on data analysis and normative comparisons are still worth exploring and discussing.

### 5.1. About the Results of Intelligent Diagnosis Using an Intelligent Assessment System

The study showed that the mathematical learning strategies of high school students in Jianzha County, Qinghai Province, were at a middle level; only the score rate of mathematical metacognitive strategies (S = 69.20%) was higher than that of the overall questionnaire (S = 68.07%), while the scoring rate of mathematical cognitive strategies, similar to mathematical resource-management strategies, was lower than that of the overall questionnaire. During the assessment process, by communicating with mathematics teachers, we found that teachers rarely emphasize the importance of mathematical learning strategies and

seldom guide students to use them in the usual teaching process, and some teachers cannot even understand how to develop students' mathematical learning strategies. By communicating with students, we found that most of them are afraid of difficulties and challenges when learning mathematics, and they cannot master the strategies and methods of learning mathematics; thus, although they have spent a long time studying mathematics, their math scores are still not ideal. It can be seen that the interviews of the students and teachers involved are consistent with the diagnosis results.

*5.2. About the Effect of Case Intervention Using an Intelligent Strategy Implementation System*

In previous studies on how to improve students' mathematics learning strategies, most researchers focus on how to improve teachers' teaching strategies [39], while few researchers focus on how to help students use mathematics learning strategies independently. This study uses an intelligent evaluation system to conduct batch evaluations for the tested classes, and automatically and efficiently output one-to-one diagnostic reports. Studies have verified that the intervention of mathematics learning strategy for students can improve their mathematics learning strategies [40], which is consistent with the intervention results of the three case students, indicating that the intelligent assessment and strategy implementation system can help students improve mathematics learning strategy levels. After three months of individual intervention, the mathematics academic performance of the three students improved. The reasons for this mainly include two aspects: on the one hand, the intervention program in the intelligent system is suitable for all high school students with different levels of mathematics learning strategies; on the other hand, the intervention program in intelligent systems is targeted. It provides one-to-one precise improvement measures according to the characteristics of different students. Because this research addresses the root problems that exist in mathematics learning strategy, it realizes the improvement in individual mathematics learning performance.

*5.3. Conclusions*

In this study, the level of mathematics learning strategies of high school students in Jianzha County, Qinghai Province, China, was investigated using a mathematics learning strategies questionnaire for high school students. The following conclusions were obtained through data analysis and results analysis. Firstly, the overall level of mathematics learning strategy of high school students in Jianzha County is at the middle level of the norm in Tianjin. The "three regions and three prefectures", as deeply impoverished areas, have achieved remarkable results in the fight against poverty in education. The students' levels in mathematical cognitive strategies, metacognitive strategies, and resource-management strategies are all at an intermediate level, and there is room for further improvement. Secondly, there are typical cases with poor results in each sub-dimension of mathematical learning strategies. Thirdly, through measuring and diagnosing the mathematics learning strategy levels of the three students using the Intelligent Assessment System, we formulated a targeted individual intervention plan for them and carried out a three-month intervention for improvement. After the intervention, the three students' math learning strategy levels improved significantly, and the effect size of the difference in the mathematics learning performance before and after the intervention was greater than the large effect size, which showed that the implementation of the intervention program could help improve the students' mathematics learning achievements.

## 6. Recommendations

Effective mathematics learning strategies can significantly improve students' mathematics performance and boost students' learning attitudes and emotions. Therefore, the key to improving mathematics performance is to evaluate the current situation of high school students' mathematics learning strategies and to make improvements based on deficiencies. The assessment tool has been used in many cities in China and has been well received by teachers and students everywhere. In addition, in the future, a norm table of high school

students' mathematics learning strategy levels can be established in each city or region. By consulting the norm table, we can comprehend the level of the students' mathematics learning strategy level and the relative position of the high school students in the city or region to understand the students' mathematics learning strategy level more accurately.

The study showed that the "Mathematics Learning Quality Intelligent Assessment and Strategy Implementation System", developed by Wang Guangming et al. and approved by the National Copyright Administration of China (NCAC), can intelligently measure mathematics learning strategies at scale and in batches for entire districts, entire schools, and individual students, and can quickly output targeted diagnostic reports for each participant while automatically providing corresponding improvement measures. Students can identify the root causes of their problems and solve their problems in mathematics learning strategies based on the improvement plans provided by the intelligent system. Therefore, schools and teachers can make full use of the intelligent system to cultivate students' mathematics learning strategies, thereby improving students' mathematics learning efficiency.

Teachers should be flexible in using various ways to instruct students in mathematics learning strategies, as well as integrating and utilizing all assessment forces and assessment resources. Based on the improvement plan provided by the intelligent system, teachers should supplement it with necessary individual interviews to further understand students' problems and current situations when using mathematics learning strategies; they should then flexibly supervise and instruct students.

First of all, it is recommended that students fully understand their learning strategy level and existing problems based on the intelligent diagnosis report. Secondly, it is necessary to deeply understand the value of mathematics learning strategies for improving the quality of mathematics learning and the necessity of using strategies. At the same time, students should follow the suggestions of intelligent systems, and continuously improve the level of mathematics learning strategies under the scientific and reasonable guidance and supervision of teachers. Through the migration and generalization of the application of learning strategies in the process of daily mathematics learning, students continuously improve their awareness of using mathematics learning strategies and realize the significance and value of using such strategies.

**Author Contributions:** Conceptualization, G.W.; methodology, G.W. and X.C.; software, G.W. and Y.K.; validation, X.C., F.W. and D.Z.; formal analysis, X.C., F.W. and D.Z.; investigation, Y.K., D.Z. and M.S.; resources, G.W.; data curation, X.C., F.W. and D.Z.; writing—original draft preparation, X.C. and D.Z.; writing—review and editing, X.C. and D.Z.; visualization, Y.K. and M.S.; supervision, G.W.; project administration, G.W.; funding acquisition, G.W. All authors have read and agreed to the published version of the manuscript.

**Funding:** This research was funded by the Key Cultivation Project of Tianjin Teaching Achievement Award: Research and Development of Mathematics Learning Assessment Tool and Its Practical Application, grant number PYJJ-036.

**Institutional Review Board Statement:** Not applicable.

**Informed Consent Statement:** Informed consent was obtained from all subjects involved in the study.

**Data Availability Statement:** The data presented in this study are available from the corresponding author upon reasonable request.

**Acknowledgments:** We thank all the high school students in Jianzha County, Huangnan Tibetan Autonomous Prefecture, Qinghai Province, China, who participated in the questionnaire research, and we thank all the teachers who helped us with the research.

**Conflicts of Interest:** The authors declare no conflict of interest.

## Appendix A

Questionnaire on Mathematical Learning Strategies of Senior High School Students in Jianzha County, Qinghai Province

Hello, classmates!

Through this math learning survey, you will be provided with targeted math learning advice to help you improve your math skills. There is no good or bad score in this questionnaire, it is only an objective evaluation of your mathematics learning situation, so please write your real answer carefully.

**Table A1.** Questionnaire on Mathematical Learning Strategies of Senior High School Students.

| | Mathematical Learning Strategies | | | Score | | |
|---|---|---|---|---|---|---|
| 1 | When I find similar mathematical concepts and formulas, I will compare their similarities and differences. For example, point-slope, oblique-intercept, intercept, and general expressions and their applicability. | 1 | 2 | 3 | 4 | 5 |
| 2 | I will compare myself to classmates with differences in learning behavior to find ways to improve math performance. | 1 | 2 | 3 | 4 | 5 |
| 3 | When learning mathematical content with more knowledge points such as functions and conic sections, I will organize its images, properties, and expressions in a table, with pictures and texts, to assist learning. | 1 | 2 | 3 | 4 | 5 |
| 4 | I will summarize and summarize the mathematical knowledge, problem-solving methods, and typical examples involved in the same type of questions. | 1 | 2 | 3 | 4 | 5 |
| 5 | When I take a math test, I always have to look at the test paper first to understand the number and type of questions, so that I know what to do. | 1 | 2 | 3 | 4 | 5 |
| 6 | Teachers ____ teach us mathematics learning methods, such as effective organization of review, setting learning goals, efficient use of time, etc. | 1 | 2 | 3 | 4 | 5 |
| 7 | I attach importance to analogy learning and variant training. For example, after learning the proof of the cosine theorem of acute triangles, I try to use this method to prove the cosine theorem of right triangles and obtuse triangles. | 1 | 2 | 3 | 4 | 5 |
| 8 | When I am learning new mathematical concepts and formulas, I am often disturbed by similar concepts and formulas. For example, due to the influence of the prism volume formula, I forget to multiply by 1/3 when calculating the pyramid volume. | 1 | 2 | 3 | 4 | 5 |
| 9 | In math class, I notice and manage my distractions. | 1 | 2 | 3 | 4 | 5 |
| 10 | I am aware of my mathematics learning status, and I am aware of the changes in my learning status. | 1 | 2 | 3 | 4 | 5 |
| 11 | When solving the problem, I will pay attention to what conditions are given in the question, what problems are required to be solved, and analyze which conditions are helpful for the problems I solve. | 1 | 2 | 3 | 4 | 5 |
| 12 | Parents tell us about math learning methods, such as: making study plans, using time efficiently, etc. | 1 | 2 | 3 | 4 | 5 |
| 13 | When I encounter a complex mathematical problem, I will break it down into several small problems and break them down. | 1 | 2 | 3 | 4 | 5 |
| 14 | I make predictions about the time it takes for math learning tasks, and guarantee them. | 1 | 2 | 3 | 4 | 5 |
| 15 | I organize the mathematics study case, homework, test papers, and other materials in high school in an orderly manner, which is convenient for review and reference. | 1 | 2 | 3 | 4 | 5 |
| 16 | When encountering math teachers with different teaching styles, I pay attention to whether my learning style is compatible with them. | 1 | 2 | 3 | 4 | 5 |
| 17 | I often find out what I have mastered by reciting and dictating mathematical concepts, formulas, and theorems. | 1 | 2 | 3 | 4 | 5 |
| 18 | I learned about mathematics learning methods by browsing the Internet, reading books, etc. | 1 | 2 | 3 | 4 | 5 |
| 19 | Before the math class, I will preview and figure out key points and difficulties, so that I can focus on listening in class. | 1 | 2 | 3 | 4 | 5 |
| 20 | When I can't solve a problem with one method, I use other methods. | 1 | 2 | 3 | 4 | 5 |

**Table A1.** *Cont.*

| | Mathematical Learning Strategies | Score | | | | |
|---|---|---|---|---|---|---|
| 21 | I don't know how to improve my math learning efficiency. | 1 | 2 | 3 | 4 | 5 |
| 22 | In high school mathematics study, I can effectively use the scattered time (such as breaks between classes, waiting time, etc.) and the whole time (such as evening self-study, weekends, etc.). | 1 | 2 | 3 | 4 | 5 |
| 23 | I resisted the evaluation of my math study by teachers and parents and rarely listened to their opinions. | 1 | 2 | 3 | 4 | 5 |
| 24 | In the math test, I will pay attention to my problem-solving speed and answering time. | 1 | 2 | 3 | 4 | 5 |
| 25 | When I study mathematics, I often focus on two uses, such as listening to music while studying. | 1 | 2 | 3 | 4 | 5 |
| 26 | I attach great importance to the results of the mathematics test and rank and use this as the basis for my evaluation of the adjustment of my mathematics learning situation. | 1 | 2 | 3 | 4 | 5 |
| 27 | When I am frustrated in the process of mathematics learning, I will review my experience of mathematics progress to improve my self-confidence in mathematics learning. | 1 | 2 | 3 | 4 | 5 |
| 28 | For the math homework, the things I did wrong and won't do in the test paper, I would put them together and revisit them after a while. | 1 | 2 | 3 | 4 | 5 |
| 29 | In math class, I control myself not to get distracted. | 1 | 2 | 3 | 4 | 5 |
| 30 | I will create a good environment for myself to study mathematics, such as maintaining the environment's quietness, neatness of the desktop, etc., to improve the efficiency of mathematics learning. | 1 | 2 | 3 | 4 | 5 |
| 31 | I have a clear math study plan. | 1 | 2 | 3 | 4 | 5 |
| 32 | After entering high school, no matter how busy I am, I can always arrange time reasonably and complete my math homework on time. | 1 | 2 | 3 | 4 | 5 |
| 33 | In the process of solving mathematical problems, I will choose the problem-solving method according to the problem type and degree of difficulty. For example, when answering multiple-choice questions, I use the substitution method and the special value method; when solving complex solid geometric problems, I use the vector method. | 1 | 2 | 3 | 4 | 5 |
| 34 | When solving math problems, I often encourage myself to think independently, not to be too unreasonable. Rely on teacher explanations or answer prompts. | 1 | 2 | 3 | 4 | 5 |
| 35 | I enrich my mathematics knowledge and cultivate my interest in mathematics by participating in high school mathematics competitions, high school mathematics summer camps, and other activities. | 1 | 2 | 3 | 4 | 5 |
| 36 | When learning solid geometry, I will use objects such as walls, rulers, books, etc. to build models. help to understand. | 1 | 2 | 3 | 4 | 5 |
| 37 | I know goal A and direct action of my mathematics learning and encourage myself to work hard for it. | 1 | 2 | 3 | 4 | 5 |
| 38 | I try to use some online resources (such as micro-courses) and mathematical software (such as Geometric Sketchpad, etc.) to assist in learning. | 1 | 2 | 3 | 4 | 5 |
| 39 | After completing my math homework, I check my answer against the correct answer Is it correct and standard? | 1 | 2 | 3 | 4 | 5 |
| 40 | When I can't remember some mathematical concepts, formulas, theorems, properties, etc., I will review the materials in time for review. | 1 | 2 | 3 | 4 | 5 |
| 41 | I often review the content of each chapter of high school mathematics that I have learned, and sort out the heavy weight of each chapter. Points, difficulties, and common question types and test sites. | 1 | 2 | 3 | 4 | 5 |
| 42 | I often summarize my approach to learning mathematics and reflect on whether it works. | 1 | 2 | 3 | 4 | 5 |
| 43 | After entering high school, I often asked my teachers and classmates for some math problems. | 1 | 2 | 3 | 4 | 5 |
| 44 | In the study of various modules of high school mathematics, I will consciously practice more for the modules that I am not good at. | 1 | 2 | 3 | 4 | 5 |

**Table A1.** *Cont.*

| | Mathematical Learning Strategies | | Score | | | |
|---|---|---|---|---|---|---|
| 45 | After finishing the math knowledge of each module, I will sort out the knowledge and form a knowledge network. | 1 | 2 | 3 | 4 | 5 |
| 46 | When I study mathematics, I will mark important content and memorize it repeatedly. | 1 | 2 | 3 | 4 | 5 |
| 47 | I will describe related concepts and theorems in various ways such as text language, graphic language, and symbolic language. | 1 | 2 | 3 | 4 | 5 |
| 48 | I always feel that the efficiency of studying mathematics is the highest only before the exam. | 1 | 2 | 3 | 4 | 5 |
| 49 | When encountering an unfamiliar math problem, I often try to relate it to a problem I am familiar with. | 1 | 2 | 3 | 4 | 5 |
| 50 | During my high school mathematics study, I was able to adjust my mentality so that I was neither too slack nor too nervous. | 1 | 2 | 3 | 4 | 5 |
| 51 | When learning new mathematics content, I will preview or read related materials in advance (such as completing a study guide or reading-related mathematics history, etc.). | 1 | 2 | 3 | 4 | 5 |
| 52 | During my high school mathematics study, I will arrange rest and study reasonably according to the activity of my brain. | 1 | 2 | 3 | 4 | 5 |
| 53 | Since high school, I have not systematically developed a math study plan. | 1 | 2 | 3 | 4 | 5 |
| 54 | I often pass a simple test after a certain stage of study to test the effectiveness of my mathematics learning. | 1 | 2 | 3 | 4 | 5 |

Note: 5 = Absolutely appropriate, 4 = Appropriate, 3 = Uncertain, 2 = Inappropriate, 1 = Absolutely inappropriate.

## Appendix B

Outline of Interview Questions for Case Student A

Dear classmate:

To further understand your situation in mathematics learning, you are invited to participate in this interview. The interview content is only used for this research. Personal information and interview content will not be disclosed without your consent and will be strictly controlled. Confidential. Therefore, I hope you can answer every question truthfully, and thank you for your support and cooperation in this research!

**Table A2.** Outline of Interview Questions.

| Number | Questions |
|---|---|
| 1 | Are parents able to provide guidance and assistance in the study and use of your mathematics learning methods? |
| 2 | Are teachers able to provide guidance and assistance in the study and use of your mathematics learning methods? |
| 3 | Please evaluate your learning methods in mathematics learning |
| 4 | Do you regularly categorize the math problems you have done? For example: Integrate mathematical problem-solving ideas and refine mathematical problem-solving methods. |
| 5 | Will you use your usual math homework and test paper feedback to conduct self-analysis and adjust your attitude and method of math learning? |
| 6 | After learning new mathematical knowledge, can you review it in time and often? |
| 7 | When you listen to the math class, will you grasp the key points and make comments on the key points? |
| 8 | Based on understanding mathematical concepts and other knowledge, will you try to explain them in your language to deepen your understanding and memory of knowledge? |

**Table A2.** *Cont.*

| Number | Questions |
|---|---|
| 9 | Will you actively use math problems to test your mastery of math knowledge? For example, when dealing with the wrong questions in the exercises, we should correct them in time, and think about "why did I make mistakes", and "have I mastered all the knowledge points of the wrong question inspection". |
| 10 | Do you try to summarize the main points and framework of the mathematical knowledge of a module or unit regularly in the form of outlines, tables, graphs, etc.? |
| 11 | Do you review math in a variety of practice formats? For example, reviewing knowledge points by asking and answering questions and doing math exercises with the same desk. |
| 12 | Do you use memory skills to help you memorize some mathematical formulas and theorems? For example: memorizing formulas. |

## Appendix C

Outline of Interview Questions for Case Student B

Dear classmate:

To further understand your situation in mathematics learning, you are invited to participate in this interview. The interview content is only used for this research. Personal information and interview content will not be disclosed without your consent and will be strictly controlled. Confidential. Therefore, I hope you can answer every question truthfully, and thank you for your support and cooperation in this research!

**Table A3.** Outline of Interview Questions.

| Number | Questions |
|---|---|
| 1 | Are parents able to provide guidance and assistance in the study and use of your mathematics learning methods? |
| 2 | Are teachers able to provide guidance and assistance in the study and use of your mathematics learning methods? |
| 3 | Please evaluate your learning methods in mathematics learning |
| 4 | Will you formulate reasonable and specific short-term and long-term goals for mathematics learning according to your current mathematics learning situation? For example: set a time to do homework. |
| 5 | Do you plan your math test time properly? |
| 6 | Do you have the habit of checking the results after completing your math homework and exam papers? For example: bring the answer to the question for verification; recalculate the question again, etc. |
| 7 | When solving math problems, do you compare your results with your life and learning experiences? For example, when learning plane geometry, the positional relationship between two straight lines, line-surface, and surface-surface relationship can be considered about the corners in life. |
| 8 | Are you able to reflect on your learning attitude and learning methods from time to time? For example: often reflect on "whether I can listen carefully in the math class" and "whether I can take the math homework assigned by the teacher seriously". |
| 9 | In general, what do you think your progress in mathematics depends on? For example effort, luck, ability? |
| 10 | Please evaluate the students around you who have good grades in mathematics. What is the reason for them to achieve good grades in mathematics? |
| 11 | Do you deepen your thinking and understanding of mathematics learning through self-questioning? For example: in mathematics learning, I often ask myself "Have I mastered all the mathematics knowledge today" and "What is the reason for not mastering all of them". |

## Appendix D

Outline of Interview Questions for Case Student C

Dear classmate:

To further understand your situation in mathematics learning, you are invited to participate in this interview. The interview content is only used for this research. Personal information and interview content will not be disclosed without your consent and will be strictly controlled. Confidential. Therefore, I hope you can answer every question truthfully, and thank you for your support and cooperation in this research!

**Table A4.** Outline of Interview Questions.

| Number | Questions |
|--------|-----------|
| 1 | Are parents able to provide guidance and assistance in the study and use of your mathematics learning methods? |
| 2 | Are teachers able to provide guidance and assistance in the study and use of your mathematics learning methods? |
| 3 | Please evaluate your learning methods in mathematics learning |
| 4 | Do you have some thoughts about math problems before seeking help from others? |
| 5 | Will you choose a suitable math learning environment? For example, choose to study math in a quiet environment with suitable light, temperature, and quiet. |
| 6 | Are you able to correctly face the negative emotions generated in mathematics learning? For example: when you have bad emotions in the process of mathematics learning, take a short rest, take a deep breath, or do some other healthy activities that you are interested in to divert your attention or experience the beauty and value of mathematics, cultivate and stimulate good mathematics learning motivation, and actively adjust, Get rid of a bad mood? |
| 7 | Have you raised your hand to ask the teacher a question when you have questions during math class? Do you choose to ask the teacher when you have questions after class? |
| 8 | Would you create an ideal math learning environment for yourself? For example, pay attention to the orderly arrangement of desks, chairs, and books; after the mathematics study of a certain module or a certain unit, regularly organize mathematics study materials such as mathematics papers, mathematics study cases, and mathematics homework books. |
| 9 | Do you make a to-do list daily and prioritize the things that need to be done? |
| 10 | When you encounter difficulties in the process of mathematics learning, will you be afraid of difficulties? If so, would you use self-motivation to increase your motivation to learn mathematics? |
| 11 | Do you have a strict time limit for completing math homework and avoid procrastination? For example: make a detailed homework schedule, the contents of the schedule include what homework to write in a certain period, how much to write, and then complete the math homework on time within the period specified in the homework plan. |

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
