# Peer review of "Development and Application of an Intelligent Assessment System for Mathematics Learning Strategy among High School Students—Take Jianzha County as an Example"

_sustainability, doi:10.3390/su141912265_

Round 1
Reviewer 1 Report
The study needs to elaborate on various aspects in order to make it clear to the readers. Look at my comments.
I did not understand the methodology until I read the entire section. So, please make it clear as to what is proposed to be done, the intervention, etc., under the research methodology.
Author Response
Thank you for your time and wisdom in reviewing the manuscript.
Thank you for your valuable comments.
To make the methodology of this paper clearer and more organized, we have added "3.4 Intelligent intervention process" in Section 3. Methodology. (on page 9, from line 353 to line 391)We explained in detail the complete process of using the Mathematics Learning Strategies Intelligent Assessment and Strategy Implementation System to perform an intelligent intervention on case students.
Reviewer 2 Report
The title of the article is good (but needs to be shortened a bit).
2. General Assessment: the content of the article is very high quality, with
a good article structure, and produces good novelty.
3. Purpose: Explain the purpose of the research clearly and usefully.
4. The research problem is clearly written, written explicitly and clearly in
the text with a theoretical framework and supported by data.
5. Literature review provides the basis for good and correct research.
Synthesis of references well. It uses a credible source. The author cites
sources accurately and reflects what was published in the original
sources, and relates and supports the theoretical framework.
6. Research Methods Explained Rationally, Clearly and Completely.
7. Results and Implications Supported by Accurate Data, with good
novelty. The author draws reasonable and data-based conclusions.
Reviewer Conclusion: This article deserves to be published and can be
continued into the production process for publication.

Author Response
Thank you for your high praise of the manuscript.
Thank you for your time and wisdom in reviewing the manuscript.
Please see the attachment.

Reviewer 3 Report
The aim of the manuscript is to present development, implementation and test of an intelligent assessment and strategy implementation system for mathematics learning strategies by integrating AI technology. The system enables intelligent evaluation and diagnosis of mathematics learning strategies of students, and intelligently prescribe "prescriptions" based on the "cause of the disease”. The system is implemented to measure the level of mathematics learning strategies of high school students.
The authors introduce the topic and based on the literature review proposed research questions for the study. The presented study is prolongation of the previous comprehensive and long-term work of the authors.
Methods and results are clearly presented, while results and conclusions are aligned with the study aims.
The manuscript is easy to follow and to comprehend the content. Used references are relevant and up to date.
Author Response

(The authors gave the same response as above.)
